# CHEMALGEBRA: ALGEBRAIC REASONING ON CHEMICAL REACTIONS

## ABSTRACT

While showing impressive performance on various kinds of *learning* tasks, it is yet unclear whether deep learning models have the ability to robustly tackle *reasoning* tasks. Measuring the robustness of reasoning in machine learning models is challenging as one needs to provide a task that cannot be easily shortcut by exploiting spurious statistical correlations in the data, while operating on complex objects and constraints. To address this issue, we propose CHEMALGEBRA, a benchmark for measuring the reasoning capabilities of deep learning models through the prediction of stoichiometrically-balanced chemical reactions. CHEMALGEBRA requires manipulating sets of complex discrete objects – molecules represented as formulas or graphs – under algebraic constraints such as the mass preservation principle. We believe that CHEMALGEBRA can serve as a useful test bed for the next generation of machine reasoning models and as a promoter of their development.

## 1 INTRODUCTION

Deep learning models, and Transformer architectures in particular, currently achieve the state-of-the-art for a number of application domains such as natural language and audio processing, computer vision, and computational chemistry (Lin et al., 2021; Khan et al., 2021; Brașoveanu & Andonie, 2020). Given enough data and enough parameters to fit, these models are able to learn intricate correlations (Brown et al., 2020). This impressive performance on machine *learning* tasks suggests that they could be suitable candidates for machine *reasoning* tasks (Helwe et al., 2021).

Reasoning is the ability to manipulate a knowledge representation into a form that is more suitable to solve a new problem (Bottou, 2014; Garcez et al., 2019). In particular, algebraic reasoning includes a set of reasoning manipulations such as abstraction, arithmetic operations, and systematic composition over complex objects. Algebraic reasoning is related to the ability of a learning system to perform *systematic generalization* (Marcus, 2003; Bahdanau et al., 2018; Sinha et al., 2019), i.e. to robustly make predictions beyond the data distribution it has been trained on. This is inherently more challenging than discovering correlations from data, as it requires the learning system to actually capture the true underlying mechanism for the specific task (Pearl, 2009; Marcus, 2018).

Lately, much attention has been put on training Transformers *to learn how to reason* (Helwe et al., 2021; Al-Negheimish et al., 2021; Storks et al., 2019; Gontier et al., 2020). This is usually done by embedding an algebraic reasoning problem in a natural language formulation. Natural language, despite its flexibility, is imprecise and prone to *shortcuts* (Geirhos et al., 2020). As a result, it is often difficult to determine whether the models' performance on reasoning tasks is genuine or it is merely due to the exploitation of spurious statistical correlations in the data. Several works in this direction suggest (Agrawal et al., 2016; Jia & Liang, 2017; Helwe et al., 2021) the latter is probably the case.

In order to effectively assess the reasoning capabilities of Transformers, we need to accurately design tasks that i) operate on complex objects ii) require algebraic reasoning to be carried out and iii) cannot be shortcut by exploiting latent correlations in the data. We identify *chemical reaction prediction* as a suitable candidate for these desiderata. First, chemical reactions can be naturally interpreted as transformations over bags of complex objects: reactant molecules are turned into product molecules by manipulating their graph structures while abiding certain constraints such as the law of mass conservation. Second, these transformations can be analysed as algebraic operations over (sub-)graphs (e.g., by observing bonds forming and dissolving (Bradshaw et al., 2019)), and balancing them to

preserve mass conservation can be formalised as solving a linear system of equations, as we will show in Section 2. Third, the language of chemical molecules and reactions is much less ambiguous than natural language and by controlling the stoichiometric coefficients, i.e., the molecule multiplicities, at training and test time we can more precisely measure systematic generalization. Lastly, Transformers already excel at learning reaction predictions (Tetko et al., 2020; Irwin et al., 2022).[1] Therefore, we think this can be a solid test bed to measure the current gap between learning and reasoning capabilities of modern deep learning models.

The main contributions of this paper are the following:

1. We cast chemical reaction prediction as a reasoning task where the learner has not only to predict a set of products but also correct stoichiometric coefficient variations (Section 2).

2. We evaluate the current state-of-the-art Transformers for chemical reaction predictions, showing that they fail to robustly generalise when reasoning on simple variants of the chemical reaction dataset they have been trained on (Section 3).

3. We introduce CHEMALGEBRA as a novel challenging benchmark for machine reasoning, in which we can more precisely measure the ability of deep learning models to algebraically reason over bags of graphs in in-, cross- and out-of-distribution settings (Section 4).

## 2  PREDICTING CHEMICAL REACTIONS AS ALGEBRAIC REASONING

To illustrate our point, let us consider the Sabatier reaction: it yields methane ($CH_4$) and water ($H_2O$) out of hydrogen ($H_2$) and carbon dioxide ($CO_2$), in the presence of nickel (Ni) as a catalyst. In chemical formulas:

$$1CO_2 + 4H_2 \xrightarrow{Ni} 1CH_4 + 2H_2O \tag{1}$$

where formulas encode complex graph structures where atoms are nodes and chemical bonds edges:

A reaction prediction learning task hence consists of outputting a bag of graphs, the *products* (right hand side), given the bag of graphs consisting of *reactants* (left hand side) and *reagents* (i.e. the catalysts, over the reaction's arrow).

The multiplicities of the molecules, also called their *stoichiometric coefficients*, express the fractional proportions of reactants to yield a certain proportion of products. For example, one needs a 4:1 ratio of hydrogen molecules and carbon dioxide to produce a 1:2 ratio of methane and water. A reaction is (mass) *balanced* when its stoichiometric coeffients are well placed such that the sum of the number of atoms for every element across products shall be the same of that across reactants, i.e., it satisfies the principle of mass conservation (Whitaker, 1975). *Unbalanced* reactions, on the other hand, would be chemically implausible.

This constraint over atoms of the molecules underpins the true chemical mechanism behind reactions: bonds between atoms break and form under certain conditions but atoms do not change. In reasoning terms, this is a symbol-manipulating process where bags of graphs are deconstructed into other bags of graphs. A machine reasoning system that would have learned this true chemical mechanism, would be able to perfectly solve the chemical reaction prediction task for all balanced reactions and for all possible variations of stoichiometric coefficients.

As humans, we can balance fairly complex chemical reactions quite easily.[2] For machines, this process can be formalised as finding a solution of a potentially undetermined system of linear equations. For example, we can write the Sabatier reaction as:

$$r_1 \cdot CO_2 + r_2 \cdot H_2 + r_3 \cdot Ni = p_1 \cdot CH_4 + p_2 \cdot H_2O + p_3 \cdot Ni \tag{2}$$

---

[1]An extended overview of the related works in chemical reaction prediction is given in Appendix A.

[2]We learn to do it from very few examples, e.g. a handful of reactions taken from chemistry textbooks in high school. Without following an explicit algorithm, we can usually perform balancing in an intuitive way, by leveraging our quick arithmetic skills to count the atoms for an element and match the numbers on both sides of the equation, iteratively changing the stoichiometric coefficients until all elements are balanced.

where the variables $r_1$, $r_2$, $r_3$, $p_1$, $p_2$, $p_3$ represent the stoichiometric coefficients of the molecules they refer to. Molecules of reagents act as a special kind of *confounders*: since they are not changed during the reaction, they must appear on both sides of the equation with the same coefficient (i.e., $r_3 = p_3$). Under this rewriting, it becomes even more evident how a full chemical reaction can be interpreted as an algebraic equation where stoichiometric coefficients are the unknown variables.

Then, we can represent the molecule of $CO_2$ with the vector $[1, 0, 2, 0]$, indicating one atom of carbon, zero hydrogens, two oxygens, and zero nickles. Analogously, $H_2$ can be encoded as $[0, 2, 0, 0]$, and so on. Therefore, Eq. (1) can be rewritten as the following linear system:

$$\begin{bmatrix} 1 & 0 & 0 \\ 0 & 2 & 0 \\ 2 & 0 & 0 \\ 0 & 0 & 1 \end{bmatrix} \begin{bmatrix} r_1 \\ r_2 \\ r_3 \end{bmatrix} = \begin{bmatrix} 1 & 0 & 0 \\ 4 & 2 & 0 \\ 0 & 1 & 0 \\ 0 & 0 & 1 \end{bmatrix} \begin{bmatrix} p_1 \\ p_2 \\ p_3 \end{bmatrix}. \tag{3}$$

It is straightforward to verify that the minimum norm solution of Eq. (3) is $\mathbf{r} = [1, 4, 1]$, $\mathbf{p} = [1, 2, 1]$, thus conforming to Eq. (1). We now devise a set of reasoning tasks exploiting this perspective.

## 2.1 "Type 1" and "Type 2" chemical reaction reasoning tasks

We propose to cast chemical reaction prediction as a reasoning task, where the model has to predict both the graphs corresponding to the product molecules (i.e. the right hand side of the reaction) and the exact multiplicities of such molecules, given a particular input consisting of reagents and reactants equipped with varying stoichiometric coefficients. This is in stark contrast with the vanilla reaction prediction learning setting: in it, models are trained on reactions without stoichiometric information, largely on unbalanced reactions (see Section 3.2); in our setting, stoichiometric coefficients are not only present but, at prediction time, can greatly differ from those the model has seen at training time. By doing so, we can better control and measure systematic generalization.

For example, given a reference reaction, we can multiply by a factor all the stoichiometric coefficients in it. For Sabatier reaction, we would obtain the following input-output pair for a factor of two:

$$2CO_2 + 8H_2 + 2Ni \quad (\text{INPUT}) \qquad 2CH_4 + 4H_2O + 2Ni \quad (\text{OUTPUT}). \tag{4}$$

In the rest of this paper, we will refer to this type of reasoning task as **Type 1** task. Alternatively, we can just add a certain number of molecules on the left hand side. Some of these additional molecules might not take part in the reaction, since there might not be enough reactants to bond with. For example, if we add two $CO_2$ and two Ni to Sabatier reaction we expect a model to predict them in the right hand side in addition to its usual outputs:

$$3\,CO_2 + 4\,H_2 + 3\,Ni \quad (\text{INPUT}) \qquad CH_4 + 2\,H_2O + 3\,Ni + 2\,CO_2 \quad (\text{OUTPUT}). \tag{5}$$

We refer to this as a **Type 2** task. Type 2 reactions are harder to reason with than Type 1, but should still be easy for a machine learning model that has learned the true underlying chemical mechanism.

There is one final aspect to consider: if the test coefficients are sampled from the same data distribution as the training coefficients, there is still a chance that the model would be able to predict them just by pattern matching. Conversely, learning the actual algebraic reasoning behind the stoichiometry of chemical reactions, e.g., by solving the linear system of Eq. (3), empowers the model to solve any stoichiometry problem, regardless of the actual values of the coefficients observed during training.

To control for these aspects, in addition to the usual **in-distribution** scenario where both training and test coefficients are sampled from the same set of integer numbers $\mathcal{S}_{in}$, we consider also an **out-of-distribution** setting where the training and test reactions are instantiated with coefficients coming from the disjoint sets of integers $\mathcal{S}_{in}$ and $\mathcal{S}_{out}$, respectively. Analogously, we can test a **cross-distribution** scenario, where the training set is divided into two halves. The reactions of the first half are instantiated with coefficients selected from $\mathcal{S}_{in}$, while the second half's coefficients are selected form $\mathcal{S}_{out}$. The cross-distribution test set will contain the same reactions of the training set but with "swapped stoichiometry": that is, the first half of the test set will have coefficients from $\mathcal{S}_{out}$, the second half from $\mathcal{S}_{in}$. We argue that a well-trained reasoning model should have no problem to semantically disentangle the stoichiometric coefficients from the molecules, thus achieving the same performance in both the in-distribution, cross-distribution, and out-of-distribution scenarios.

Table 1: Statistics of the training, validation and test splits of the USPTO dataset before and after our rebalancing. Percentages in parenthesis are w.r.t. the original dataset.

|  | TOTAL | BALANCED (%) | RE-BALANCED (%) | USPTO-BAL (%) |
|---|---|---|---|---|
| TRAINING SET | 409035 | 18815 (4.59) | 162220 (39.66) | 181035 (44.26) |
| VALIDATION SET | 30000 | 1292 (4.31) | 11868 (39.56) | 13160 (43.87) |
| TEST SET | 40000 | 1809 (4.52) | 15973 (39.93) | 17782 (44.45) |

## 3 CAN TRANSFORMERS PERFORM ALGEBRAIC REASONING?

In this section, we question whether current deep learning models can effectively solve the algebraic reasoning tasks induced by the stoichiometry of chemical reactions, as discussed in Section 2. In order to do so, we first need to build a suitable dataset of chemical reactions that can be processed by state-of-the-art models for reaction predictions. We then evaluate these models on some meaningful variations of the task and discuss their limitations.

### 3.1 HOW TO BUILD A BENCHMARK OF BALANCED REACTIONS

A natural candidate is the USPTO-MIT dataset (Lowe, 2012), a large collection of chemical reactions extracted from US patents from 1976 to 2016 and represented in the SMILES format (Weininger, 1988). SMILES reactions are text strings encoding a linearization of the molecular graphs of each molecule participating in the reaction.[3] We employ the version popularized by Jin et al. (2017), often referred to as USPTO-MIT. This version contains a subset of polished reactions, obtained by removing duplicate and incorrect ones. For simplicity, in the rest of this paper we will refer to "USPTO-MIT" as simply "USPTO". This dataset is composed of 409k training reactions, 30k validation reactions and 40k test reactions (see Table 1).

Unfortunately, reactions in the USPTO dataset are mostly unbalanced: we found that less than 4.6% of the reactions in the dataset are mass balanced and readily usable for our tasks. For the remaining reactions, in many cases only the major product of the reaction was recorded, while disregarding minor byproducts, such as $H_2O$ or HCl, probably not deemed interesting from a patent perspective. Examples of these reactions are illustrated in Fig. 1. While this does not impact patents, it can affect the generalization capability of the learned models, as we will show next.

Fortunately, the missing byproducts can be sometimes deduced from the unbalanced reaction: if the set of missing atoms of one side of the reaction is enough to unambiguously reconstruct one (or more) valid molecules on the other side, we add such reconstructed molecules to that side of the reaction. The rightmost column in Table 1 shows the number of reactions contained in the balanced subset of USPTO that we used as the basis for the augmentation procedures described in Section 4. Using this strategy, we were able to re-balance about 44% of the original reactions for the training, validation and test sets. We denote this re-balanced version of USPTO as USPTO-BAL and use it next.

### 3.2 TYPE 1 AND TYPE 2 VARIANTS OF USPTO

We build two variants of the original USPTO dataset, called USPTO-T1 and USPTO-T2, as instances of the Type 1 and Type 2 reasoning tasks introduced in Section 2.1. For T1, we multiply all coefficients by two, in practice duplicating the molecule representations in the SMILES string of a reaction (as the SMILES format does not allow to represent stoichiometric coefficients). For T2 we add a randomly-selected molecule to the reactants. The detailed procedure is discussed in Appendix B.

These two variants should not, in theory, pose significant challenges for chemical reaction models that learned the true reaction mechanisms of USPTO, since despite shifting from the original distribution, they still use the same molecules and reaction mechanisms of USPTO.

In the case of USPTO-T1, the reactants contain all the required additional molecules to perform the reaction twice, so the output should correspond to the same multiple of the original products. This is even easier considering that the original reactions in USPTO generally involve a single molecule per

---

[3]Note that SMILES strings represent hydrogen implicitly, when the resulting molecule is not ambiguous.

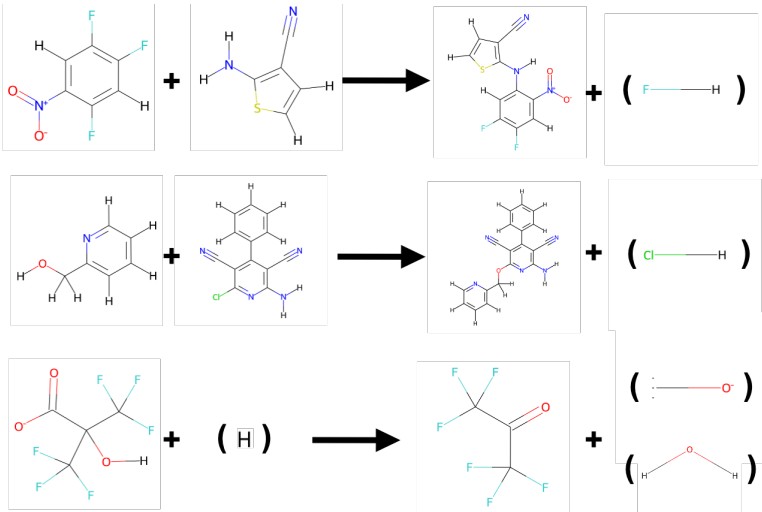

Figure 1: Examples of re-balanced reaction of the USPTO-BAL dataset. The inferred byproducts are shown between brackets (the reagents have been omitted for simplicity).

Table 2: Results of state-of-the-art Transformers for reaction prediction on the USPTO dataset and the BAL, T1 and T2 variations. We report the top-1 accuracy (ACC) for all models. We also report the at-least-one (ALO) accuracy for the T1 and T2 variants. All metrics are reported as percentages.

| STATE-OF-THE-ART MODEL | USPTO ACC | USPTO-BAL ACC | USPTO-T1 ACC | USPTO-T1 ALO | USPTO-T2 ACC | USPTO-T2 ALO |
|---|---|---|---|---|---|---|
| MOLTRANS. (SCHWALLER ET AL., 2019) | 90.4 | 1.39 | 0.04 | 56.66 | 0.03 | 28.32 |
| CHEMFORMER (IRWIN ET AL., 2022) | 92.8 | $< 10^{-2}$ | 0.23 | 34.41 | $< 10^{-2}$ | 2.61 |
| G2S (DGCN) (TU & COLEY, 2021) | 90.3 | 1.37 | $< 10^{-2}$ | 83.02 | $< 10^{-2}$ | 40.79 |
| G2S (DGAT) (TU & COLEY, 2021) | 90.3 | 1.40 | $< 10^{-2}$ | 84.21 | $< 10^{-2}$ | 41.43 |

type. On the other hand, in the case of USPTO-T2, the added molecule is not sufficient to trigger multiple reactions, so it should just be copied in output by the model. As we will show in the next section, in practice *this is not the case*.

## 3.3 CURRENT CHEMICAL MODELS ARE ACTUALLY ALCHEMICAL MODELS

We now evaluate the systematic generalization of state-of-the-art models for reaction prediction trained on USPTO on the chemically-sound variation and the reasoning variants introduced in Section 3.1 and Section 3.2. We focus on Transformer-based language models that operate on SMILES representations (Schwaller et al., 2019; Irwin et al., 2022) as well Transformers that employ graph neural networks (GNNs) to parse the molecules structures and be permutation invariant (Tu & Coley, 2021). For additional details about these models, we refer the reader to Appendix B.

For all models, we report the top-1 prediction accuracy (ACC) on the BAL, T1 and T2 dataset variants as well as the original USPTO. On the T1 and T2 variants, a correct prediction would involve the model output both the right molecular graph and the right multiplicity of each product molecule. This proved extremely challenging for all models. As such, we also report the "at-least-one" accuracy (ALO) score which consider the prediction correct if at least one of the ground truth molecules is being predicted, despite its multiplicity being inaccurate.

The results in Table 2 are striking: while all models are able to reach over 90% of top-1 accuracy over the standard unbalanced USPTO test set, the performance drops sharply below 2% on USPTO-T1 and USPTO-T2 but also on USPTO-BAL, despite this variant only adds a small number of byproduct molecules. This fall in performance is also evident for the ALO metric.

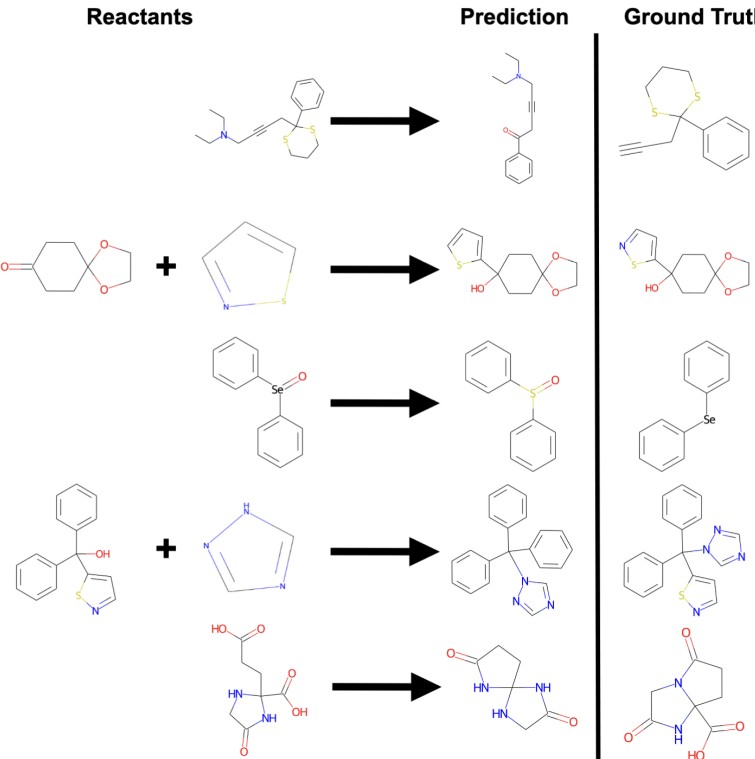

Figure 2: **Examples of alchemical reactions** as predicted by state-of-the-art Transformer models on the original USPTO dataset. (see Table 2 and Appendix B) where atoms of chemical elements appear or disappear between reactants and predicted products regardless of the mass preservation principle. For example, in the first reaction, sulphur (S) is turned into oxygen (O), and in the third selenium (Se) into sulphur. In the second and fourth, nitrogen (N) and sulphur disappear as elements from the products, while in the last one only two atoms of oxygen out of four remain.

The distribution shifts in the original training distribution introduced in our variants highlight that these state-of-the-art models are not learning the true chemical mechanism, but a *brittle* function approximation that leverages statistical correlations present in the data. This is somehow expected as these models have been trained on unbalanced reactions and their predicted products often do not satisfy the mass preservation constraint. In fact, predicted product molecules might contain not only an unbalanced number of atoms per element, but also introduce atoms of elements that were not present in the reactants and reagents or making other elements disappear from the products. We show some qualitative examples of these "alchemical" predictions reported on the original USPTO dataset,[4] in Fig. 2.

So, why are current state-of-the-art models not able to perform basic forms of reasoning over chemical data? We think there are several reasons: part of the problem might reside in the training data itself, not containing chemically plausible reactions. Datasets based on USPTO, in their current form, are not able to pose a significant learning challenge that is also sound from a chemical point of view. Another part of the problem resides in the intrinsic limitations of deep learning architectures that are currently used for this kind of tasks. While efficient and flexible, these models are too keen on leveraging statistical patterns of the data, and have a hard time to perform out-of-distribution predictions even on simple variations of the original task, thus failing to achieve systematic generalisation.

One could argue that, if we added the Type 1 and Type 2 reaction variations in a novel training set, the models learned on it would be robust to them. This is not a satisfying answer for models that should systematically generalize, as we could always test their performance on a new batch of reaction

---

[4]In nuclear reactions the appearance or disappearance of elements would be allowed, but this kind of reactions is out of the scope of USPTO and our work.

Table 3: Example of the CHEMALGEBRA input-output variations obtained from the same initial reference reaction and represented in our stoichiometrically-augmented SMILES and raw chemical formulas (FORMULA) languages.

| | | REACTANTS | PRODUCTS |
|---|---|---|---|
| | ORIGINAL | $Cl^- + HCl + C_6H_{12}O + C_8H_{10}$ | $HCl + Cl^- + C_{14}H_{22}O$ |
| T1 | SMILES | {3}Cl.{3}[Cl-].
{3}CCCCCC1CO1.{3}Cc1cccc(C)c1 | {3}[Cl-].{3}Cl.
{3}CCCCC(CO)c1ccc(C)cc1C |
| | FORMULA | {3}HCl.{3}Cl-.
{3}C6H12O.{3}C8H10 | {3}HCl.{3}Cl-.
{3}C14H22O |
| T2 | SMILES | {3}Cl.{5}[Cl-].
{5}Cc1cccc(C)c1.{3}CCCCC1CO1.
{2}CCCCC(CO)c1ccc(C)cc1C | {3}Cl.{5}[Cl-].
{3}Cc1cccc(C)c1.{1}CCCCC1CO1 |
| | FORMULA | {3}HCl.{4}Cl-.
{1}C14H22O.{1}C6H12O.{2}C8H10 | {3}HCl.{4}Cl-.
{3}C8H10.{2}C6H12O |

variants involving slightly different stoichiometric coefficient manipulations, e.g., multiplying them by three instead of two. Designing a suitable benchmark suite to systematically train and evaluate models over several variations of these manipulations poses certain challenges in terms of input specification and controlling potential shortcutting. We discuss and overcome these challenges in the next section.

## 4 THE CHEMALGEBRA BENCHMARK

Building on the previous considerations, we now design the CHEMALGEBRA benchmark. Starting from the balanced variant of USPTO, described in Section 3 (USPTO-BAL), we devise eight tasks for assessing the reasoning capabilities of deep learning models. First of all, we need to find a way to explicitly represent the stoichiometric coefficients directly (as duplicating SMILES sub-strings becomes impractical for long molecules and large coefficients numbers, which are needed for a challenging benchmark). To this end, we define a **stoichiometrically-augmented SMILES string** language, where each coefficient is encoded as an integer surrounded by curly braces preceding each molecule in the reaction. For example, Sabatier reaction from Eq. (1) becomes

$$\{1\}O=C=O.\{4\}[HH].\{1\}[Ni] > > \{1\}C.\{2\}O.\{1\}[Ni] \qquad (6)$$

in this new SMILES dialect. To better understand how the reasoning task of predicting the correct reaction mechanisms affects the performance, we devise a second, simpler, representation where molecules in the reactions are encoded as **raw chemical formulas**. Sabatier reaction encoded in this way is:

$$\{1\}CO2.\{4\}H2.\{1\}Ni > > \{1\}CH4.\{2\}H2O.\{1\}Ni \quad . \qquad (7)$$

Note that in both cases the reagent Ni is copied explicitly on both sides of the reaction. These alternative notations in CHEMALGEBRA pose different challenges for deep learning models. The SMILES representation is more verbose and lower-level, allowing for a complete reconstruction of the original molecular structure. However, SMILES strings can be ambiguous (Arús-Pous et al., 2019), with more than one string representing the same molecule.[5] They also do not explicitly represent Hydrogen atoms, so the models would need to extract that implicit information by its own and properly balance it. The formula notation, on the other hand, is more compact and more abstract. It represents all the atoms, and their multiplicities, in an explicit way, but the correct reaction mechanism is harder to predict, as the formula does not contain structural information.

We can now instantiate the two reasoning tasks defined in Section 2.1 with the new notation for stoichiometric coefficients in CHEMALGEBRA. For **Type 1** reactions, we randomly select a single multiplication factor from a set and apply it to all molecules in the reaction. Since the reactions in USPTO-BAL are balanced, but rarely contain molecules with coefficients greater than one, Type 1 reactions will likely contain molecules sharing the same random stoichiometric coefficient. We

---

[5]We use the canonized molecular string representation (Arús-Pous et al., 2019).

employ this variant to precisely measure how much the learning model can shortcut the reaction reasoning task by just predicting the same coefficient of the input reagents for the output products.

Instead, for **Type 2** reactions, each molecule in the reaction reagents, reactants and products is randomly assigned a different coefficient and allowed to appear on the left or right side in addition to its original placement, thus creating confounder molecules that might not react. This process requires taking some precautions in order to ensure that the resulting reaction is still balanced. We detail the complete procedure to create Type 2 reactions in Appendix C. As this would make it difficult for a model to shortcut coefficient prediction, Type 2 reactions pose a greater challenge than Type 1. Table 3 illustrates examples of Type 1 and 2 derivations from a reference reaction, encoded in stoichiometric SMILES and raw formula representations.

We now have to decide how many times shall we create Type 1 and 2 variants of a reaction from USPTO-BAL at training time. We control this factor of variation by devising two augmentation strategies to measure how different training sizes could impact generalization:

**x1 augmentations** : each reaction of USPTO-BAL is used to create a single stoichiometric reaction in CHEMALGEBRA.

**x5 augmentations** : each reaction of USPTO-BAL is used to create five stoichiometric reactions in CHEMALGEBRA. All the stoichiometric reactions contain the same input/output molecules, but differ by the sampled values of the stoichiometric coefficients.

This leaves us with a total of 8 datasets variants, one for each combination of the 2 reasoning types, 2 molecule encodings and 2 augmentation amounts. A comprehensive overview of the eight CHEMALGEBRA datasets is included in Appendix G. Lastly, we need to determine the different sets of coefficients to sample from across train and test distributions. We thus instantiate the chemical reactions in CHEMALGEBRA by sampling the stoichiometric coefficients in the interval $\mathcal{S}_{in} = [1, 5]$. We retain the same train, validation and test split as the original USPTO-BAL dataset. As outlined in Section 2.1, in order to test systematic generalisation, we build three different test sets for each dataset variant:

**In-distribution** test set: the test reactions are instantiated with stoichiometric coefficients drawn in the interval $\mathcal{S}_{in}$. This is the "easy" scenario where the model should be expected to use the shortcuts for Type 1 reasoning.

**Out-of-distribution** test set: the test reactions are instantiated with stoichiometric coefficients drawn in the $\mathcal{S}_{out} = [6, 10]$ interval. This is the "hard" prediction task where to succeed a model would also require the arithmetic knowledge to derive coefficients in $\mathcal{S}_{out}$ from those in $\mathcal{S}_{in}$. This challenging setting is introduced explicitly to stimulate future research to inject external knowledge or ways to bind symbols to concepts.

**Cross-distribution** test set: half of training and test reactions is instantiated with stoichiometric coefficients in $\mathcal{S}_{in}$, the other half in $\mathcal{S}_{out}$. This is a "medium" difficulty scenario where the model can infer arithmetic knowledge from the training data and still shortcut predictions for Type 1 reactions.

## 4.1 BASELINE EXPERIMENTS

As part of our benchmark, we provide reference baseline results for the eight variations of CHEMAL-GEBRA described in Section 4. We employ a Transformer architecture[6] akin to the state-of-the-art models of Table 2 (see Appendix D for the details about the architecture and the training procedure). Our aim with these baselines is to analyse the performance variations with respect to different molecule representations, coefficient assignment types, and dataset sizes.

We measure the performance of the model as in a multilabel classification setting, where the task is to predict not only the exact bag of product molecules but also their stoichiometric coefficients.[7] To do

---

[6]We chose to focus on the Transformer for our baseline because i) the models with the highest accuracy on forward reaction prediction are all based on Transformers, ii) Transformers are already currently used for many types of reasoning tasks (Helwe et al., 2021) and iii) while other specialized approaches, such as MEGAN (Sacha et al., 2021), can bake simple chemical rules into the model by design, they cannot be used to solve different kinds of reasoning tasks.

[7]We measure the "molecule-only" performance in Appendix F.

Table 4: Exact match (EM), Jaccard (JAC) and F1 scores evaluating the prediction of both molecules and coefficients on the 8 variations of CHEMALGEBRA for in-, cross- and out-of-distribution generalization.

| | Type | Aug. | In | | | Cross | | | Out | | |
|---|---|---|---|---|---|---|---|---|---|---|---|
| | | | EM | JAC | F1 | EM | JAC | F1 | EM | JAC | F1 |
| FORMULA | T1 | x1 | 29.99 | 63.03 | 72.42 | 37.81 | 67.40 | 75.72 | 0.36 | 3.89 | 6.48 |
| | | x5 | 28.87 | 62.10 | 71.69 | 25.85 | 60.64 | 70.59 | 0.38 | 7.30 | 12.15 |
| | T2 | x1 | 1.10 | 39.87 | 53.53 | 0.03 | 31.43 | 44.84 | 0.00 | 13.71 | 23.05 |
| | | x5 | 1.30 | 41.01 | 54.71 | 0.06 | 32.57 | 45.97 | 0.00 | 13.29 | 22.37 |
| SMILES | T1 | x1 | 4.17 | 44.43 | 57.49 | 5.96 | 45.74 | 58.55 | 0.04 | 1.88 | 3.18 |
| | | x5 | 2.62 | 41.86 | 54.99 | 2.77 | 42.01 | 55.18 | 0.08 | 1.87 | 3.38 |
| | T2 | x1 | 0.03 | 23.37 | 35.37 | $< 10^{-2}$ | 22.73 | 34.57 | 0.00 | 8.90 | 15.61 |
| | | x5 | 0.04 | 24.00 | 36.18 | $< 10^{-2}$ | 22.73 | 34.67 | 0.00 | 7.44 | 13.08 |

so, we consider the bag of labels to be the bag of product molecules, in which a molecule appears as many times as its correct coefficient states. We therefore employ classical multi-label performance metrics (Tsoumakas & Katakis, 2007) for the top-1 generated bags of products. We use the exact match (EM) score , the multi-label accuracy, also called Jaccard (JAC) score, and the F1 score metrics (see Appendix E for details). These metrics are differently forgiving, with the EM score being the most strict and F1 the most permissive.

The results in Table 4 show some interesting trends. First, as expected, the EM scores drops sharply on the out-of-distribution setting. Not only the model is not able to recover the exact multiplicity of product molecules, but also their presence, as the F1 and JAC scores drop to much less than half the in-distribution scores. Type 2 tasks are confirmed to be much harder than Type 1, as highlighted especially by EM scores.

Second, the baseline can actually achieve better out-of-distribution JAC and F1 scores Type 2 than Type 1 tasks. This can be explained by considering that JAC and F1 scores are more permissive than EM and can increase if the prediction contains more instances of one molecule as labels, even if the complete configuration does not exactly match.

Third, augmenting the training data fivefold (x5) does not yield any breakthrough performance improvement. On the contrary, for Type 1 reactions, it seems that additional augmentations can slightly confuse the model on which shortcut to use. Overall, this suggests that the CHEMALGEBRA tasks are challenging and robust to the usual data augmentation techniques that are usually used to improve the performance of Transformer-based models (Schwaller et al., 2019; Irwin et al., 2022). We recommend running CHEMALGEBRA x1 if one needs to save computation.

Finally, balancing raw chemical formulas is reported to be easier than doing that on SMILES. It seems that predicting the stoichiometric coefficients is easier when using formulas, while even Type 1 in-distribution reaction predictions are challenging when using graph structures, encoded as SMILES, as inputs.

## 5 CONCLUSION

Our capacity to measure the reasoning capabilities for current machine learning models is bounded by the benchmark tasks that we have at our disposal now. Many of the current datasets used for this purpose cast reasoning as a subroutine of more complex problems, often expressed in natural language (Helwe et al., 2021; Wei et al., 2022). By doing so it becomes hard to disentangle the systematic generalization ability of a model from a number of confounding factors such as mere statistical correlations in the data (Zhang et al., 2022; Branco et al., 2021).

We introduced CHEMALGEBRA to have a solid benchmark for robustly measuring the reasoning abilities of deep learning models over complex objects, such as bags of graphs. CHEMALGEBRA offers a more controlled and versatile experimental setting – including different reasoning tasks and graph encodings – while remaining highly challenging in both the in- and out-of-distribution

settings. In fact, it requires to combine statistical learning over graphs with algebraic reasoning over the (sub-)graph structures and their multiplicity. As such, we think that the CHEMALGEBRA task can serve as a useful test bed for the upcoming generation of machine reasoning models.

## 5.1 LIMITATIONS AND FUTURE DIRECTIONS

While completely disentangling learning and reasoning might not be possible (as it is not in many real-world applications), we stress that the intended use of CHEMALGEBRA is mainly for the evaluation of the reasoning abilities of neural models and specifically Transformers. Therefore, it should not be considered to be a dataset to train real-world models for pure chemical reaction prediction. We remark that other neural reaction prediction, such as MEGAN, which inject prior knowledge about chemical reaction predictions, might possess a more favourable inductive bias to solve the reasoning task. Our focus is however on Transformers, which are emerging as general-purpose learning and reasoning machines. We believe CHEMALGEBRA can be a solid benchmark to measure their capabilities and offer a more controlled environment where shortcutting can be limited.

We recognize the limitations of the re-balancing technique introduced in Section 3, which is a simple heuristic and may abstracts-away several chemical nuances that can be relevant for pure chemical reaction predictions. At the same time, we believe that can it be a simple initial solution to propel more rigorous and challenging algebraic reasoning benchmarks. Type 2 augmentation, while sound from an algebraic point of view, might also introduce noise in the chemical reaction prediction task.

We plan to design additional variations of CHEMALGEBRA to take into account the different facets of machine and human reasoning as well as more chemically and physically plausible constraints such as modeling redox reactions.

## REPRODUCIBILITY STATEMENT

The dataset files, code to run the experiments scripts to compute the results of this paper can be found at `https://anonymous.4open.science/r/ChemAlgebra`. The construction of the BAL, T1 and T2 variants of USPTO is described in Section 3 of the main paper. Additional details about the USPTO variants and the state-of-the-art models are reported in Appendix B. Section 4 describes the eight dataset variants included in CHEMALGEBRA. The detailed algorithm used to sample the stoichiometric coefficient of Type 2 tasks is reported in Appendix C. Appendix D contains additional information about the architecture and training procedure of the Transformer model used to compute the baseline results of Section 4.1. An extended description of the multilabel metrics used for evaluation is included in Appendix E. Finally, an extensive overview of the CHEMALGEBRA dataset variants is given in Appendix G.

## ACKNOWLEDGMENTS

We wish to thank Adriàn Javaloy and Frank Mollica for their precious discussions and suggestions, which were crucial to improve the quality of the final manuscript.

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

## A    RELATED WORK

**ML for chemical predictions.**    Computational methods for chemistry have a long history of applications. While the first methods for chemical reaction predictions relied mostly on hand-crafted symbolic rules (Salatin & Jorgensen, 1980), in recent years the computational the community interests shifted towards the use of deep learning techniques and derivatives. Nowadays, deep learning is used for a big variety of chemically related tasks, such as molecular graph generations (Sousa et al., 2021; Goyal et al., 2020; Bacciu & Podda, 2021; Simonovsky & Komodakis, 2018; Mercado et al., 2021), molecule optimisation (Jin et al., 2018; Griffiths & Hernández-Lobato, 2020; Korovina et al., 2020), and chemical reaction predictions. Focusing on the latter, we can find two related approaches: forward prediction, where the goal is to infer the product molecules given a set of reactants, and retrosynthesis, a specular problem to forward synthesis consisting into predict the original reactants given a final product.

**DL for chemical predictions.**    Initial attempts to use use deep learning models for reaction prediction used neural network to first identify the reaction centre in the reactants molecules (Coley et al., 2019) or pairwise reactivity of atom bonds (Jin et al., 2017; Fooshee et al., 2018). The proposals of the models were then used to algorithmically construct the final product of the reaction. A relevant collection of works aim at generating the whole reaction trees by iteratively selecting predefined reaction templates (Bradshaw et al., 2018; Segler et al., 2018; Bradshaw et al., 2020; Gao et al., 2021; Dai Nguyen & Tsuda, 2021; Nguyen & Tsuda, 2022). Reaction templates are helpful for preventing the models to generate incorrect molecules that violates chemical rules, however they tend to be too restrictive in general, making it more difficult for the models to generalise to new types of reactions.

In the attempt to overcome the limitation of fixed reaction templates, it was proposed to combine powerful Transformer-based (Vaswani et al., 2017) language models with the text SMILES representations of molecules in order to generate reaction outcomes in one step. These models leverage the flexibility of their architecture to directly learn a mapping between the reactants and the products. The first work of this kind is the Molecular Transformer (Schwaller et al., 2019), able to beat the previous state of the art on both forward and backward prediction. Further improvements quickly followed with the Augmented Transformer model (Tetko et al., 2020). In this work, the authors focused on augmenting the original dataset by computing different variations of SMILES strings representing the same molecule.

The current top-1 accuracy state of the art for text-based, template-free models is represented by the Chemformer (Irwin et al., 2022), where the authors use a self-supervised pre-training on a large variety of known molecules in order to increase performance on the reaction prediction downstream task. An interesting line of research tries to leverage the natural graph structure of molecules. Tu & Coley (2021) combine Transformers with Deep Graph Networks (Bacciu et al., 2020) to get better initial molecular embeddings, while (Tavakoli et al., 2022) build an hypergraph representation for the reaction by augmenting the disconnected molecular graph with "hypernodes" that represent molecules and reactions sides as a whole.

Finally, an orthogonal approach to chemical reaction prediction tries to leverage natural chemical constraints for ensuring the generation of chemically-plausible reactions. Bradshaw et al. (2018) model low-level reaction mechanisms for linear electron flow heterolythic reactions. This model generate the final products by predicting the linear electron flow on the initial set of reactants. Qian et al. (2020) combine neural networks with integer level programming constraints for expressing simple chemical constraints. Other works try to predict simplified version of the actual reactions mechanisms (often called "pseudo-mechanisms"), either via auto-regressively generating edits to the molecular graphs (Sacha et al., 2021) or single-step predicting all possible bonds formation and deletion using a multi-pointer decoding network (Bi et al., 2021). The architectural constraints of these models generally ensure that the products resulting from a reaction are sound from a chemical perspective.

**Benchmarking and studying logical and algebraic reasoning.**    After the recent impressive results of large scale pre-trained Transformers, several works started to investigate the reasoning abilities of these models (Geirhos et al., 2020; Helwe et al., 2021; Tran et al., 2021). Wang et al. (2021) show that large scale language models can only generalise well when the test distribution is the same of the training distribution, while they struggle in cross-distribution and out-of-distributions scenarios. Liu

et al. (2020) introduce a dataset based on natural language multiple-choice questions. Each question requires a certain amount of logical reasoning in order to reach the correct answer. The baseline results show that the performance of current language models is still far behind human beings. The out-of-distribution problem is also highlighted in Razeghi et al. (2022), showing that the accuracy of these models on a certain training item is proportion to the number of times that that item as been seen in during training.

# B    ADDITIONAL RESULTS AND DETAILS FROM SECTION 3

The USPTO-T1 and USPTO-T2 variations are built in the following way:

- **USPTO-T1**: we duplicate all the molecules on their respective sides of the reaction (double reactants, and double products).
- **USPTO-T2**: for each reaction, we randomly select a molecule from either the reactants or the products. We then replicate the selected molecule on both sides of the reaction.

Please note that, with USPTO data, there is no a straightforward way to represent stoichiometry: in order to represent a double molecule in the reaction, we explicitly need to copy it twice in the data. Due to architectural limitations of the considered models, we are constrained to perform only a limited amount of augmentations (e.g. it would be impossible to replicate the entire reactants more than twice without exceeding the maximum sequence length). Table 5 report, as an example, the Sabatier reaction re-adapted to the BAL, T1 and T2 variants.

For the experiments, we consider the following Transformer-based state-of-the-art models:

- Molecular Transformer (MOL.T) (Schwaller et al., 2019): the first transformer-based model that framed the problem of reaction predictions as sequence-to-sequence text translation.
- Augmented Transformer (AUG.T) (Tetko et al., 2020): extension of the Molecular Transformer, it was trained using different data augmentation schemes in order to boost its performance.
- Chemformer (CHEMF) (Irwin et al., 2022): it yields the current state of the art of top-1 accuracy on the reaction prediction tasks based on USPTO.
- Graph2SMILES (G2S) (Tu & Coley, 2021): a Transformer-based model that uses a graph-based neural encoder for learning more expressive representations of the input molecules. The authors of G2S presents two versions (*dgcn* and *dgat*) of the model employing a different type of neural encoder.

Tables 6, 7, 8 ad 9 report the detailed results of these model over the USPTO, USPTO-BAL, USPTO-T1 and USPTO-T2 variants. We compute the following metrics:

- *Validity rate* (VAL): ratio of predictions that have a correct molecular structure.
- *Accuracy* (ACC): ratio of correct predictions.
- *At-least-one-accuracy* (ALO): we define a prediction to be "at-least-one accurate" if it contains all the ground truth molecules at least once. It can therefore be seens as a "molecule-only" accuracy, disregarding the multiplicity of the molecules.
- *Balanced predictions rate* (BAL): ratio of predictions that are balanced (i.e. that contains the same atoms as in the reactants).
- *Deficitary predictions rate* (DEF): ratio of predictions that do not contain atoms than are present in the reactants.
- *Exceeding predictions rate* (EXC): ratio of predictions that contain atoms that are not present in the reactants.
- *Deficitary and exceeding prediction rate* (D+E): ratio of prediction that are both exceeding and deficitary.

The table shows that, while the number of valid and accurate predictions is over 90% for all models, the balanced predictions are just a minority, less than 10%. Most of the unbalanced predictions are of the deficitary type, while a relative minority is of the exceeding type. One of the causes for the bad performance can be identified in the training data: as shown in Sec. 3.1, many of the reactions contained in USPTO are already unbalanced. This unbalance is reflected on the outputs of these models, another sign that these models are relying on memorising the data rather than trying to actually retrieve the correct underlying chemical mechanisms.

Table 5: Example of the the Sabatier reaction adapted to fit the BAL, T1 and T2 variations of the original USPTO. In T1, the reactants and products are doubled. In T2, one molecule ($CO_2$) is added on both side of the reaction.

|  | Reactants | Products |
|---|---|---|
| Original | $CO_2 + H_2$ | $CH_4 + H_2O$ |
| BAL | $CO_2 + H_2 + H_2 + H_2 + H_2 + Ni$ | $CH_4 + H_2O + H_2O + Ni$ |
| T1 | $CO_2 + CO_2 + H_2 + H_2 + Ni$ | $CH_4 + CH_4 + H_2O + H_2O + Ni$ |
| T2 | $CO_2 + CO_2 + H_2 + Ni$ | $CO_2 + CH_4 + H_2O + Ni$ |

Table 6: Results of predictions of state-of-the-art models on the USPTO dataset.

|  | VAL | ACC | BAL | DEF | EXC | D+E |
|---|---|---|---|---|---|---|
| MOL.T | 100.0 | 90.4 | 9.06 | 90.86 | 0.28 | 0.21 |
| AUG.T | 99.97 | 91.1 | 9.44 | 90.40 | 0.59 | 0.43 |
| CHEMF | 87.20 | 92.8 | 6.71 | 76.33 | 48.28 | 31.33 |
| Graph2SMILES (dgcn) | 99.98 | 90.3 | 8.83 | 90.62 | 1.60 | 1.06 |
| Graph2SMILES (dgat) | 99.98 | 90.3 | 8.88 | 90.52 | 1.67 | 1.08 |

Table 7: Results on predictions of state-of-the-art models on USPTO-BAL.

|  | VAL | ACC | BAL | DEF | EXC | D+E |
|---|---|---|---|---|---|---|
| MOL.T | 99.85 | 1.39 | 1.50 | 98.48 | 0.30 | 0.29 |
| CHEMF | 78.20 | 0.0 | 4.51 | 92.39 | 22.52 | 19.43 |
| G2S (dgat) | 99.89 | 1.37 | 1.48 | 98.50 | 0.15 | 0.14 |
| G2S (dgcn) | 99.85 | 1.40 | 1.46 | 98.53 | 0.16 | 0.15 |

Table 8: Results on predictions of state-of-the-art models on USPTO-T1 dataset.

|  | VAL | ACC | ALO | BAL | DEF | EXC | D+E |
|---|---|---|---|---|---|---|---|
| MOL.T | 99.03 | 0.04 | 56.66 | 0.18 | 99.76 | 1.41 | 1.37 |
| CHEMF | 91.88 | 0.23 | 34.41 | 0.29 | 99.22 | 6.31 | 5.83 |
| G2S (dgat) | 99.60 | 0.0 | 83.02 | 0.01 | 99.97 | 0.67 | 0.66 |
| G2S (dgcn) | 99.67 | 0.0 | 84.21 | 0.01 | 99.97 | 0.64 | 0.64 |

Table 9: Results of predictions of state-of-the-art models on USPTO-T2 dataset.

|  | VAL | ACC | ALO | BAL | DEF | EXC | D+E |
|---|---|---|---|---|---|---|---|
| MOL.T | 99.03 | 0.03 | 28.32 | 0.38 | 99.08 | 5.76 | 5.23 |
| CHEMF | 67.42 | 0.0 | 2.61 | 2.48 | 90.00 | 25.06 | 17.55 |
| G2S (dgat) | 99.66 | 0.005 | 40.79 | 0.06 | 99.88 | 1.35 | 1.30 |
| G2S (dgcn) | 99.71 | 0.0025 | 41.43 | 0.04 | 99.90 | 1.19 | 1.14 |

# C  COMPLETE ALGORITHM FOR BUILDING THE TYPE-2 STOICHIOMETRIC AUGMENTATIONS

Let $\mathcal{M}$ be the set of molecules in the reaction, we first sample as before a random coefficient $k_m \in [1, 5]$ for each molecule $m \in \mathcal{M}$. We then compute

$$\hat{k} = \min_{m \in \mathcal{M}} k_m. \tag{8}$$

The final coefficients are assigned as follows: on the reactants side of the reaction, we have

$$\forall m \in \mathcal{M}. \quad k_m = \begin{cases} k_m & \text{if } m \text{ is a reactant or a reagent,} \\ (k_m - \hat{k}) & \text{if m is a product, and } (k_m - \hat{k}) > 0 \,, \\ 0 & \text{otherwise.} \end{cases} \tag{9}$$

Similarly, on the products side, we have

$$\forall m \in \mathcal{M}. \quad k_m = \begin{cases} k_m & \text{if } m \text{ is a product or a reagent,} \\ (k_m - \hat{k}) & \text{if m is a reactant, and } (k_m - \hat{k}) > 0 \,, \\ 0 & \text{otherwise.} \end{cases} \tag{10}$$

This assignment strategy takes into account the fact that specific quantities of reactants and reagents molecules are needed in order to get the respective products. The excess amount of molecules is just copied on the other side (a specular argument can be done for products). Please not that, following this algorithm, the new added molecules are chosen among the molecules that are already present in the original reaction.

## D    ADDITIONAL DETAILS ABOUT THE MODEL AND TRAINING PROCEDURE

We train a Transformer model (Vaswani et al., 2017) with 6 encoder and 6 decoder layers. The number of attention heads is set to 8. Both the embeddings size, the internal model size and the size of the fully connected layers is set to 512. Loss function used is the cross-entropy between the predicted and the target strings.

We use a batch size of 64 for all experiments. the Adam optimizer is used for training, with an initial learning rate of $10^{-5}$. The learning rate is halved when no loss improvement over the validation set is observed for 1000 steps, while the training is stopped after no loss improvement over the validation set is observed for 5000 steps. We use label smoothing with smoothing parameter $0.1$.

For evaluation, we auto-regressively generate the model's prediction one character at a time, stopping the generation either when the model predicts an "end sequence" token or after 200 timesteps. The model is implemented and trained using the *pytorch_lightning* library.

# E    MULTILABEL METRICS FOR EVALUATION

In CHEMALGEBRA, the model needs to predict a bag of product molecules, given an initial bag of reactants. This problem can be framed as a multilabel classification task, since multiple ground truth molecules (and their stoichiometric coefficients) have to be predicted at the same time. The computation of the usual classification metrics (such as accuracy, precision, recall, etc.) in the multilabel setting is not trivial, as we need to take into account different failure modes of the model at the same time. For example, the model could either predict the wrong molecule, or the right molecule but with a higher/lower coefficient than expected. For our tasks, we consider the Exact Match (EM), Jaccard (JAC) and F1 scores as multilabel metrics:

$$\text{EM} = \frac{1}{N} \sum_{i=1}^{N} \mathbf{I}(y_i = \hat{y}_i), \quad \text{JAC} = \frac{1}{N} \sum_{i=1}^{N} \frac{|y_i \cap \hat{y}_i|}{|y_i \cup \hat{y}_i|}, \quad \text{F1} = \frac{1}{N} \sum_{i=1}^{N} \frac{2|y_i \cap \hat{y}_i|}{|y_i| + |\hat{y}_i|}, \quad (11)$$

where $y_i$ and $\hat{y}_i$ are the k-hot binarized vectors of the ground truth and the prediction, respectively. $N$ is the number of test samples and $\mathbf{I}$ is the indicator variable checking for exact equality between $y_i$ and $\hat{y}_i$.

Due to computational efficiency reasons, we do not explicitly binarize the vectors. Instead, we compute the number of true positives, false positives and false negatives by comparing the coefficients of corresponding molecules in both the prediction and the ground truth.

For example, if the model predicts $3\,H_2O + 2\,HCl + CO_2$ while the ground truth is $2\,H_2O + 2\,HCl + CH_4$, we count 4 true positives (two $H_2O$ and two HCl), 2 false positives (one excess $H_2O$ and one $CO_2$), and 1 false negative (the $CH_4$ in the ground truth). These counts are then combined to get the final multilabel metrics.

# F  ADDITIONAL BASELINE RESULTS (MOLECULE-ONLY ACCURACY)

Table 10 contains the baseline results of the CHEMALGEBRA benchmarks, considering only the molecular-level accuracy (i.e. taking only into account the prediction of the correct molecules, disregarding stoichiometric coefficients). Using the same example of Appendix E, where the model predicts $3\,H_2O + 2\,HCl + CO_2$ while the ground truth is $2\,H_2O + 2\,HCl + CH_4$, we now count 2 true positives (the $H_2O$ and HCl molecules), 1 false positives ($CO_2$), and 1 false negative (the $CH_4$ in the ground truth). In practice, here we are measuring only the predicted molecular structures, disregarding the stoichiometric coefficients.

We can observe that, for Type 1 tasks, the performance are very similar to Table 4. This is not surprising, as the model can easily learn to copy the same coefficient to the output, leaving only the task of molecular graph prediction to be learned. On the other hand, the performance on Type 2 variants are higher than Table 4. This can be explained as the main challenge of Type 2 tasks is the prediction of the correct coefficients, which is disregarded by these "molecule-only" metrics. The consequence of the different choice of metrics can be also observed in the out-of-distribution performance, that in Table 10 is consistently higher than Table 4: not considering the stoichiometric coefficients makes the out-of-distribution task very similar to a standard reaction prediction tasks (with the out-of-distribution coefficients acting as confounding tokens).

Table 10: Jaccard (JAC) and F1-score (F1) on the CHEMALGEBRA datasets.

|              | test-in | | test-cross | | test-out | |
| --- | --- | --- | --- | --- | --- | --- |
|              | JAC | F1 | JAC | F1 | JAC | F1 |
| FORMULA_T1x1 | 63.03 | 72.43 | 67.39 | 75.71 | 23.96 | 35.79 |
| FORMULA_T1x5 | 62.09 | 71.69 | 60.64 | 70.59 | 26.03 | 38.11 |
| FORMULA_T2x1 | 48.89 | 61.48 | 35.10 | 48.44 | 26.35 | 39.27 |
| FORMULA_T2x5 | 50.33 | 62.80 | 36.29 | 49.54 | 27.25 | 40.26 |
| SMILES_T1x1  | 44.43 | 57.49 | 45.74 | 58.54 | 24.44 | 36.52 |
| SMILES_T1x5  | 41.85 | 54.99 | 42.00 | 55.17 | 24.50 | 36.28 |
| SMILES_T2x1  | 27.63 | 40.57 | 24.64 | 36.97 | 18.03 | 28.45 |
| SMILES_T2x5  | 28.47 | 41.54 | 24.56 | 36.98 | 14.59 | 23.50 |

## G  OVERVIEW OF THE CHEMALGEBRA VARIANTS

Table 11 contains an overview of the different CHEMALGEBRA variants that are described in Section 4. Each variant is contained in a separated folder. The filenames are given according to the following conventions:

- Files with prefix "src" contain the input molecules, while files with prefix "tgt" contain the target molecules.
- Substrings "train", "valid" or "test" refer to the respective subsets of data to be used during cross-validation.
- Test files have the additional suffixes "_in" or "_out", referring respectively to the in-distribution test set and out-of-distribution test set, as described in Section 4.
- Files with the "_cross" suffix refer to the cross-distribution setting described in Section 4. Note that the "_cross" tasks have their own dedicated training and validation sets.

In order to provide a visual intuition of the actual data contained in CHEMALGEBRA, we show in Tables 12, 13, 14, 15, 16, 17, 18 and 19 the first five training reactions for each CHEMALGEBRA variant.

Table 11: Summary of all eight variants of CHEMALGEBRA. "Name" is the name of the variant in the dataset's files. The "Representation" column specifies which representation is used for the reactions of the dataset. The "Task Type" column determines the choice of coefficients (according to the reasoning tasks defined in Section 2.1). The "Augmentation" column specifies how many stoichiometric assignments are sampled for each original USPTO reaction in the training set.

| Name | Representation | Task Type | Augmentations |
|------|----------------|-----------|---------------|
| ChemAlgebra_F_T1_x1 | Chemical formula | Type 1 | x1 |
| ChemAlgebra_F_T1_x5 | Chemical formula | Type 1 | x5 |
| ChemAlgebra_F_T2_x1 | Chemical formula | Type 2 | x1 |
| ChemAlgebra_F_T2_x5 | Chemical formula | Type 2 | x5 |
| ChemAlgebra_S_T1_x1 | SMILES string | Type 1 | x1 |
| ChemAlgebra_S_T1_x5 | SMILES string | Type 1 | x5 |
| ChemAlgebra_S_T2_x1 | SMILES string | Type 2 | x1 |
| ChemAlgebra_S_T2_x5 | SMILES string | Type 2 | x5 |

Table 12: Sample of training chemical reactions of `ChemAlgebra_F_T1_x1`.

| INPUT | TARGET |
|---|---|
| {5}C7H4ClNO4.{5}H2O.{5}CH5N | {5}HCl.{5}H2O.{5}C8H8N2O4 |
| {5}C7H7Cl2N.{5}C11H5Cl2N3S | {5}HCl.{5}C18H11Cl3N4S |
| {1}H.{1}HCl.{1}H2O.{1}C15H10ClFN4.{1}C2H4O2 | {1}C2H4O2.{1}HCl.{1}CN.{1}H2O.{1}C14H11ClFN3 |
| {4}C4H8O.{4}H.{4}C26H33N3O3.{4}Na+.{4}Na+.{4}H2O4S.{4}CH2O.{4}CHO3-.{4}H4B- | {4}C27H35N3O3.{4}O3-.{4}C4H8O.{4}Na+.{4}H4B-.{4}H2O4S.{4}Na+.{4}CH2O |
| {4}C5H5N.{4}C5H4ClNO2.{4}HCl.{4}C6H4ClNO | {4}C5H5N.{4}HCl.{4}C11H7ClN2O3.{4}HCl |

Table 13: Sample of training chemical reactions of `ChemAlgebra_F_T1_x5`.

| INPUT | TARGET |
|---|---|
| {3}C7H4ClNO4.{3}CH5N.{3}H2O | {3}HCl.{3}C8H8N2O4.{3}H2O |
| {3}CH5N.{3}C7H4ClNO4.{3}H2O | {3}H2O.{3}HCl.{3}C8H8N2O4 |
| {3}H2O.{3}C7H4ClNO4.{3}CH5N | {3}C8H8N2O4.{3}H2O.{3}HCl |
| {2}H2O.{2}C7H4ClNO4.{2}CH5N | {2}C8H8N2O4.{2}HCl.{2}H2O |
| {3}C7H4ClNO4.{3}CH5N.{3}H2O | {3}HCl.{3}C8H8N2O4.{3}H2O |

Table 14: Sample of training chemical reactions of `ChemAlgebra_F_T2_x1`.

| INPUT | TARGET |
|---|---|
| {4}C7H4ClNO4.{2}C8H8N2O4.{2}H2O.{3}CH5N | {2}H2O.{3}C8H8N2O4.{3}C7H4ClNO4.{2}CH5N.{1}HCl |
| {5}C11H5Cl2N3S.{1}HCl.{4}C7H7Cl2N.{1}C18H11Cl3N4S | {2}HCl.{4}C11H5Cl2N3S.{2}C18H11Cl3N4S.{3}C7H7Cl2N |
| {2}H2O.{3}H.{2}C15H10ClFN4.{3}CN.{2}C2H4O2.{3}C14H11ClFN3.{5}HCl | {2}H.{4}CN.{2}C2H4O2.{2}H2O.{1}C15H10ClFN4.{5}HCl.{4}C14H11ClFN3 |
| {3}C26H33N3O3.{3}CHO3-.{3}H2O4S.{1}C4H8O.{5}Na+.{5}Na+.{4}H.{4}C27H35N3O3.{1}CH2O | {1}C4H8O.{1}O3-.{5}Na+.{1}CH2O.{3}H2O4S.{2}CHO3-.{5}Na+.{5}H4B-.{2}C26H33N3O3.{3}H.{5}C27H35N3O3 |
| {2}C5H4ClNO2.{3}C5H5N.{3}HCl.{1}C6H4ClNO.{3}C11H7ClN2O3.{4}HCl | {4}C11H7ClN2O3.{3}C5H5N.{1}C5H4ClNO2.{4}HCl.{4}HCl |

Table 15: Sample of training chemical reactions of `ChemAlgebra_F_T2_x5`.

| INPUT | TARGET |
|---|---|
| {1}C8H8N2O4.{4}H2O.{2}C7H4ClNO4.{2}CH5N.{2}HCl | {4}H2O.{4}HCl.{3}C8H8N2O4 |
| {3}C8H8N2O4.{2}C7H4ClNO4.{3}H2O.{2}HCl.{5}CH5N | {3}CH5N.{5}C8H8N2O4.{4}HCl.{3}H2O |
| {3}C7H4ClNO4.{5}CH5N.{2}H2O.{4}C8H8N2O4 | {2}C7H4ClNO4.{1}HCl.{4}CH5N.{2}H2O.{5}C8H8N2O4 |
| {2}CH5N.{1}C7H4ClNO4.{3}HCl.{2}C8H8N2O4.{5}H2O | {1}CH5N.{3}C8H8N2O4.{5}H2O.{4}HCl |
| {1}H2O.{2}CH5N.{1}HCl.{1}C7H4ClNO4 | {1}H2O.{1}C8H8N2O4.{1}CH5N.{2}HCl |

Table 16: Sample of training chemical reactions of `ChemAlgebra_S_T1_x1`.

| INPUT | TARGET |
|---|---|
| `{4}CN.4O=C(O)c1ccc(Cl)c([N+](=O)[O-])c1.{4}O` | `{4}CNc1ccc(C(=O)O)cc1[N+](=O)[O-].{4}Cl.{4}O` |
| `{1}NCc1ccc(Cl)c(Cl)c1.{1}Clc1cc2c(Cl)nc(-c3ccncc3)nc2s1` | `{1}Cl.{1}Clc1cc2c(NCc3ccc(Cl)c(Cl)c3)nc(-c3ccncc3)nc2s1` |
| `{4}Cc1c(Cl)nnc(C(C#N)c2ccc(F)c(C#N)c2)c1C.{4}Cl.{4}O.`
`{4}CC(=O)O.{4}[H]` | `{4}O.{4}Cl.{4}Cc1c(Cl)nnc(Cc2ccc(F)c(C#N)c2)c1C.`
`{4}[C][N].{4}CC(=O)O` |
| `{5}[BH4-].{5}O=S(=O)(O)O.{5}C=O.{5}C1CCOC1.`
`{5}[Na+].{5}O=C([O-])O.{5}[Na+].`
`{5}CC(C)(C)OC(=O)c1ccc(NCC2(O)CCN(CCc3ccc(C#N)cc3)CC2)cc1.`
`{5}[H]` | `{5}O=S(=O)(O)O.{5}[Na+].{5}C=O.{5}[BH4-].`
`{5}CN(CC1(O)CCN(CCc2ccc(C#N)cc2)CC1)c1ccc(C(=O)OC(C)(C)C)cc1.`
`{5}[Na+].{5}[O]O[O-].{5}C1CCOC1` |
| `{3}O=c1cc(O)c(Cl)c[nH]1.{3}Cl.{3}O=C(Cl)c1cccnc1.`
`{3}c1ccncc1` | `{3}O=C(Oc1cc(=O)[nH]cc1Cl)c1cccnc1.{3}Cl.`
`{3}c1ccncc1.{3}Cl` |

Table 17: Sample of training chemical reactions of `ChemAlgebra_S_T1_x5`.

| INPUT | TARGET |
|---|---|
| `{4}O.{4}CN.{4}O=C(O)c1ccc(Cl)c([N+](=O)[O-])c1` | `{4}O.{4}CNc1ccc(C(=O)O)cc1[N+](=O)[O-].{4}Cl` |
| `{2}O=C(O)c1ccc(Cl)c([N+](=O)[O-])c1.{2}CN.{2}O` | `{2}CNc1ccc(C(=O)O)cc1[N+](=O)[O-].{2}Cl.{2}O` |
| `{4}CN.{4}O=C(O)c1ccc(Cl)c([N+](=O)[O-])c1.{4}O` | `{4}O.{4}CNc1ccc(C(=O)O)cc1[N+](=O)[O-].{4}Cl` |
| `{5}O=C(O)c1ccc(Cl)c([N+](=O)[O-])c1.{5}O.{5}CN` | `{5}Cl.{5}CNc1ccc(C(=O)O)cc1[N+](=O)[O-].{5}O` |
| `{3}O.{3}O=C(O)c1ccc(Cl)c([N+](=O)[O-])c1.{3}CN` | `{3}CNc1ccc(C(=O)O)cc1[N+](=O)[O-].{3}O.{3}Cl` |

Table 18: Sample of training chemical reactions of `ChemAlgebra_S_T2_x1`.

| INPUT | TARGET |
|---|---|
| `{3}CNc1ccc(C(=O)O)cc1[N+](=O)[O-].`
`{5}CN.{4}O=C(O)c1ccc(Cl)c([N+](=O)[O-])c1.{1}Cl.{3}O` | `{4}CNc1ccc(C(=O)O)cc1[N+](=O)[O-].`
`{3}O=C(O)c1ccc(Cl)c([N+](=O)[O-])c1.{4}CN.{3}O.{2}Cl` |
| `{4}NCc1ccc(Cl)c(Cl)c1.1Cl.`
`{3}Clc1cc2c(Cl)nc(-c3ccncc3)nc2s1` | `{2}Cl.1Clc1cc2c(NCc3ccc(Cl)c(Cl)c3)nc(-c3ccncc3)nc2s1.`
`{3}NCc1ccc(Cl)c(Cl)c1.2Clc1cc2c(Cl)nc(-c3ccncc3)nc2s1` |
| `{1}[H].1Cc1c(Cl)nnc(Cc2ccc(F)c(C#N)c2)c1C.`
`{3}Cc1c(Cl)nnc(C(C#N)c2ccc(F)c(C#N)c2)c1C.`
`{1}O.{5}CC(=O)O.{1}Cl` | `{1}Cl.{5}CC(=O)O.1O.1[C][N].`
`{2}Cc1c(Cl)nnc(Cc2ccc(F)c(C#N)c2)c1C.`
`{2}Cc1c(Cl)nnc(C(C#N)c2ccc(F)c(C#N)c2)c1C` |
| `{4}[H].{3}[Na+].{2}O=S(=O)(O)O.{5}C1CCOC1.{1}C=O.{3}[Na+].`
`{4}CC(C)(C)OC(=O)c1ccc(NCC2(O)CCN(CCc3ccc(C#N)cc3)CC2)cc1.`
`{2}O=C([O-])O.{2}[BH4-]` | `{2}[BH4-].{3}[Na+].{1}C=O.{2}O=S(=O)(O)O.1[O]O[O-].`
`{3}[H].1O=C([O-])O.{5}C1CCOC1.`
`{1}CN(CC1(O)CCN(CCc2ccc(C#N)cc2)CC1)c1ccc(C(=O)OC(C)(C)C)cc1.`
`{3}CC(C)(C)OC(=O)c1ccc(NCC2(O)CCN(CCc3ccc(C#N)cc3)CC2)cc1.`
`{3}[Na+]` |
| `{2}c1ccncc1.{3}O=C(Cl)c1cccnc1.{3}Cl.{4}Cl.`
`{4}O=C(Oc1cc(=O)[nH]cc1Cl)c1cccnc1.{1}O=c1cc(O)c(Cl)c[nH]1` | `{2}c1ccncc1.{2}O=C(Cl)c1cccnc1.{4}Cl.{4}Cl.`
`{5}O=C(Oc1cc(=O)[nH]cc1Cl)c1cccnc1` |

Table 19: Sample of training chemical reactions of `ChemAlgebra_S_T2_x5`.

| INPUT | TARGET |
|---|---|
| `{3}CN.{5}O=C(O)c1ccc(Cl)c([N+](=O)[O-])c1.`
`{3}CNc1ccc(C(=O)O)cc1[N+](=O)[O-].{4}O` | `{2}CN.{4}CNc1ccc(C(=O)O)cc1[N+](=O)[O-].`
`{4}O=C(O)c1ccc(Cl)c([N+](=O)[O-])c1.{1}Cl.{4}O` |
| `{1}CNc1ccc(C(=O)O)cc1[N+](=O)[O-].`
`{5}CN.{5}O=C(O)c1ccc(Cl)c([N+](=O)[O-])c1.{5}O` | `{2}CNc1ccc(C(=O)O)cc1[N+](=O)[O-].{4}CN.{1}Cl.`
`{5}O.{4}O=C(O)c1ccc(Cl)c([N+](=O)[O-])c1` |
| `{5}O.{2}Cl.{2}CN.{4}O=C(O)c1ccc(Cl)c([N+](=O)[O-])c1.`
`{1}CNc1ccc(C(=O)O)cc1[N+](=O)[O-]` | `{3}O=C(O)c1ccc(Cl)c([N+](=O)[O-])c1.{1}CN.`
`{3}Cl.{2}CNc1ccc(C(=O)O)cc1[N+](=O)[O-].{5}O` |
| `{1}CN.{4}CNc1ccc(C(=O)O)cc1[N+](=O)[O-].`
`{3}O=C(O)c1ccc(Cl)c([N+](=O)[O-])c1.{3}O` | `{1}Cl.{3}O.{5}CNc1ccc(C(=O)O)cc1[N+](=O)[O-].`
`{2}O=C(O)c1ccc(Cl)c([N+](=O)[O-])c1` |
| `{2}O.{5}O=C(O)c1ccc(Cl)c([N+](=O)[O-])c1.{5}CN.{2}Cl` | `{1}CNc1ccc(C(=O)O)cc1[N+](=O)[O-].{2}O.{3}Cl.`
`{4}CN.{4}O=C(O)c1ccc(Cl)c([N+](=O)[O-])c1` |

