# OpenReview forum: "ChemAlgebra : Algebraic Reasoning on Chemical Reactions"
_ICLR.cc/2023/Conference — Submitted to ICLR 2023_

### Official Review · Reviewer_Mbs4 · 2022-10-19

**Confidence:** 2
**Clarity, Quality, Novelty And Reproducibility:** This paper is clear, novel and can be…
**Correctness:** 3
**Technical Novelty And Significance:** 3
**Empirical Novelty And Significance:** 3
**Recommendation:** 5

**Strength And Weaknesses:**

Strength:
1. This dataset is somewhat interesting and novel. It is related to the problem of retro-synthesis.
2. This paper evaluate the current state-of-the-art Transformers for chemical reaction predictions, showing that they fail to robustly generalise when reasoning on simple variants of the chemical reaction dataset.

Weaknesses:
1. This paper claims that chemical reaction prediction is a reasoning task. Why? Please define the reasoning task.
2. I am not sure the role of this dataset. Did it could improve the reasoning capacity of neural networks?
3. Suppose that we design a perfect for this dataset, what can this model be used for? The applications of this dataset are still not clear for me.



**Summary Of The Paper:**

This paper introduce a new dataset named CHEMALGEBRA, in order to robustly measure the reasoning abilities of deep learning models over complex objects, such as bags of graphs. CHEMALGEBRA offers a more controlled and versatile experimental setting – including different reasoning tasks and graph encodings – while remaining highly challenging in both the in- and out-of-distribution settings. In fact, it requires to combine statistical learning over graphs with algebraic reasoning over the (sub-)graph structures and their multiplicity.

**Summary Of The Review:**

The idea of this paper is interesting but seems useless.

---

> ### Author Response · Authors · 2022-11-14
> **Response to reviewer Mbs4**
>
> We thank the reviewer for their comments and for recognizing the novelty of our benchmark and its rigorous empirical evaluation.
>
> > This paper claims that chemical reaction prediction is a reasoning task. Why? Please define the reasoning task.
>
> We define reasoning as the process of manipulating a set of symbols to yield out a task-specific computation, as usually defined in e.g., Bottou 2014 and Marcus 2003.
> In our case, the task is to transform a bag of graphs (reactant molecules) into another bag of graphs (product molecules) such that i) the products are chemically meaningful and more importantly for our work ii) the multiplicities of the elements (symbols in the molecules) in the reactants are preserved in the products. This entails the ability to perform algebraic reasoning as we can abstract point ii) into solving a linear system over the corresponding symbols (see Section 2)
>
> > I am not sure the role of this dataset.
>
> The impact of our contribution has to be understood in terms of the current benchmarks used to measure machine reasoning capabilities. Benchmarks using NLP are prone to many ambiguities coming from natural language and offering statistical shortcuts (please see our answer to reviewer 3JTU).
> Benchmarks from the neural-symbolic community are too simplistic from a reasoning perspective and involve simple tasks such as performing addition over the class of images. Please see also our answer to reviewer gwmV.
> With ChemAlgebra we raised the bar in terms of difficulty of the reasoning task and rigor in the experimental setting by controlling for the task complexity and possible shortcuts.
>
> > Suppose that we design a perfect for this dataset, what can this model be used for? The applications of this dataset are still not clear for me.
>
> The usefulness of our benchmark has not to be found in an immediate real-world application.
>  In the same way that a model perfectly solving MNIST addition (our answer to reviewer gwmV) does not aim to improve calculators.
> Algebraic reasoning is a very useful, and challenging, type of reasoning, and having models able to do it can constitute an unprecedented step forward in the field of learning/reasoning integration in neural networks applied to real-world problems.
>
>
> That being said, a model able to perfectly solve the ChemAlgebra tasks (thus showing systematic generalization capabilities) would be a model able to correctly perform algebraic reasoning over complex graph representations.
>
>  At the moment, to the best of our knowledge,  there are no models that can do that consistently.
> Second, this benchmark can be a useful tool for neural chemical reaction prediction models, for evaluating their ability to produce chemical reactions that respect some basic chemical constraints, such as the conservation of mass principle.

---

> ### Author Response · Authors · 2022-11-18
> **We hope we clarified all the reviewer's concerns**
>
> We hope that our response was able to clarify all the concerns raised by the reviewer. We kindly ask the reviewer to let us know if they have some additional questions that they would like to be addressed.
>
> We look forward to engage in a fruitful discussion!

---

> > ### Comment · Reviewer_Mbs4 · 2022-11-27
> > **Thanks for your efforts**
> >
> > After reading all the reviews and the responses, I am still not convinced that the proposed task is meaningful for either real applications or the theory of artificial intelligence. Actually, the authors also did not provide a reasonable role in real-world applications. Therefore, I have updated my evaluation to a weak reject.

---

### Official Review · Reviewer_3JTU · 2022-10-24

**Confidence:** 3
**Correctness:** 3
**Technical Novelty And Significance:** 2
**Empirical Novelty And Significance:** 2
**Recommendation:** 5

**Clarity, Quality, Novelty And Reproducibility:**

**CLARITY:** The paper is well-written throughout and is structured well. That being said, it still takes a couple of pages to finally see how the inputs and outputs look like for the chemical reaction prediction task. This can easily be remedied by either adding a short paragraph in the introduction, or perhaps adding a figure early on that demonstrates how input-output pairs look like in ChemAlgebra.
  * **Potential typo:**  The first paragraph of Section 3.1 appear to refer to two types of USPTO-MIT dataset, one that cites Lowe (2012), the another that cites Jin et al. (2017). Just wanted to check whether this is a typo and the first one is meant to be just called USPTO?

**QUALITY:** My assessment of quality hinges upon the answers to some of the clarification questions I've asked above.

**REPRODUCIBILITY**: There doesn't seem to be enough details to retrain the models used to report the main performance results. Also, the ChemAlgebra dataset also don't appear to be linked in the submission (is there a way to look at the samples during review?)

**Strength And Weaknesses:**

**STRENGTHS**:
* **Central premise is sound:** Prediction of stoichiometrically-balanced chemical reactions do indeed appear to have a strong reasoning component (manipulation of complex and structured objects, and operating under algebraic constraints). It also seems like achieving perfect accuracy on this task does require actually capturing the correct algorithm as opposed to a handful of heuristics. This makes a well-executed dataset a valuable contribution for research on improving reasoning with deep learning methods.
* **Sensible seed dataset:** The equations used in the dataset are originally collected from US Patents, which I think is a great way to ensure that the dataset is "natural" and possibly practically relevant.
* **Potentially immune to chain-of-thought reasoning:** It appears that this task is challenging enough that even the largest and most capable models would suffer at tackling it via using chain-of-thought reasoning.

****


**WEAKNESSES** (some of these concerns might be proven wrong by the authors):
* **Potential non-uniqueness of answers**: How common is it in the dataset that a given set of reactants can produce different products? I believe this could happen, especially when dealing with organic compounds (i.e. $C_5H_{12} + C_9H_{20}$ could produce $2C_7H_{16}$ or $C_3H_{8} + C_{11}H_{24}$, at least this algebraically checks out). How often does this happen in the dataset? If it does happen, what determines what the label is amongst all possible answers?
* **SOTA appear not to have been evaluated on ChemAlgebra:** The authors argue that the reported results on ChemAlgebra use a transformer architecture "akin to the state of the art models of Table 2" (Table 2 contains results on experiments presented prior to the introduction to the ChemAlgebra dataset). The details in Appendix suggest that this model is a standard (i.e. the original Vaswani et. al. version) 6 layer encoder-decoder transformer. The models used to produce Table 2 appear to either be augmented with a GNN, or pretrained with a lot of data using the BART objective. Hence, it appears that the baseline used on ChemFormer seem weaker. (It's possible that I'm missing something here that the authors can rectify.)
* **Absence of a version suitable for being used with large language models:** Current day DL-based reasoning research heavily relies on large models trained with huge pretraining datasets. Therefore, this submission would have been stronger if it had an "inference only" split that tests if large language models like GPT-3 (which are accessible for inference) can attain nontrivial accuracy. Providing explicit natural language instructions, expliting chain-of-thought reasoning and/or few-shot learning would all be interesting.
* **The Type 1 and Type 2 corruptions possibly alter the labels:** It appear that both type 1 and type 2 corruptions could lead to a change in the labels. Here are some examples I can think of:
  * Type 1: Take the following equation that shows a molecule of sugar being burnt: "C6H12O6+6O2→6CO2+6H2O". Doubling the coefficients on the left hand side, one could obtain the labels "12CO2+12H2O", or "6CO2+6H2O + C6H12O6 + 6O2" (the latter corresponds to only one of the sugar molecules being burnt, and the other remaining intact). Is one of the labels incorrect? Why?
  * Type 2 (This is the corruption used in section 3.2): Consider the salt equation "HCL + NaOH -> NaCl + H2O". Add NH4OH on the left hand side to obtain "HCL + NaOH + NH4OH". Since NH4Cl is a viable molecule, the label to this input could both be "NaCl + H2O + NH4OH" or "NH4Cl + HCL + H2O". Is one of the labels incorrect? Why?
* **Training details missing:** Details on how the models used to produce (including tuning details) Table 2 and 4 appear to be missing. (Most of the details for Table 4 are provided, though there still remain key design choices, like the loss function used)
* **The OOD split of T1 predictions seem a bit problematic:** If I'm not mistaken, the ChemAlgebra dataset doesn't need any prior pretraining. Hence, the models need to learn "what each digit means" just by working with the ChemAlgebra dataset. It looks like when the OOD split of the ChemAlgebra dataset introduces significantly larger digits (in-dist 1-5 and ood: 6-10), the model is likely presented with tokens it has not even seen before. Isn't it expected that the performance will be terrible in this split, as the models doesn't have enough training signal to learn what the OOD digits mean? I don't see an easy way around it (i.e. as the authors argue, mixing in the OOD digits makes the split in-distribution.)

****

**QUESTIONS TO THE AUTHORS:**
* **Why use a separate dataset (not ChemAlgebra) for Table 2?**
  * I was wondering why ChemAlgebra appears to be distinct from the dataset used in Section 3. Why is the evaluation on that dataset much more comprehensive than the evalution on ChemAlgebra?
* **How rule-based is the chemistry knowledge needed to perform perfectly?**
  * In other words, how would a novice human approach this task? While picking coefficients appear to be quite easy for a human, predicting what the output chemicals will be appear to be much more challenging. Is there a systematic way of approaching this?
* **Is natural language prone to shortcuts:** In the 3rd paragraph of the Introduction, there's the sentence "Natural language, despite its flexibility, is[...] prone to shortcuts." Do you mind giving an example for what a shortcut for natural language would be?
* **Loss function:** What loss function did you use to train the transformer on ChemAlgebra?
* **Soundness of the procedure to miss omitted molecules:** Is it guaranteed that the procedure used in Section 3.1 to balance unbalanced equations (by constructing one or more molecules using the excess/yet-unbalanced atoms) sound? Is it possible that this will lead to chemically unsound reactions, even though the molecules + coefficients check out?
* **Tokenization:** How are the equations tokenized?

**Summary Of The Paper:**

The authors propose the ChemAlgebra dataset, a benchmark to evaluate the reasoning capabilities of deep learning models. The task of predicting the products and their multiplicities serves as a useful testbed for evaluating reasoning, as it involves not only manipulating complex discrete objects (reactants and products can be thoughts of bags of graphs), but also acting under precise algebraic constraints induced by the mass preservation principle.

The contributions of the authors include:
* Casting chemical reaction prediction as a reasoning task,
* Putting together the ChemAlgebra dataset,
* Evaluating the performance of the vanilla transformer architecture on ChemAlgrebra, and a couple other more sophisticated models on a similar dataset, demonstrating that prediction of stoichiometrically-balanced chemical reactions is not yet a solved problem.


**Summary Of The Review:**

The idea of using chemical reaction prediction to evaluate reasoning is a great one in my opinion. It is especially timely due to its difficulty yet amenability to rule-based solutions. I imagine current-day models (even large language models) would not be able to tackle this task very well. A well-executed dataset would serve well to community now and in the future, and this submission seems to be well on the track to achieve this.

This being said, I have a number concerns in the current version of the paper that motivate current score. I'll be revising my assessment based on author feedback.

---

> ### Author Response · Authors · 2022-11-14
> **Response to reviewer 3JTU (1/3)**
>
> We thank the reviewer for having clearly summarized what we believe to be the main key contributions of our work and for praising the important direction we are taking.
>
> > How common is it in the dataset that a given set of reactants can produce different products?
>
> Note that not every algebraically (syntactically) balanced reaction corresponds to a chemically meaningful reaction. We base the ChemAlgebra tasks on the USPTO dataset that only contains chemically meaningful (but potentially unbalanced) reactions. Our rebalancing technique does not introduce ambiguities.
>
> > SOTA appear not to have been evaluated on ChemAlgebra:
>
> We remark that all SOTA models in Table 2 cannot take as input our stoichiometrically-augmented SMILE language and require molecule to be replicated in the normal SMILE representation. We experimented with copying more than twice the molecules but observed these Transformers architectures to perform even worse on ChemAlgebra or not being able to train at all. This is likely due to the fact that the input sequence length becomes too long too soon.
> Our baseline model is essentially the Molecular Transformer by Schwaller et al., which is a standard Transformer with no additional pre-processing. Also the other models in Table 2 are still, fundamentally, quite standard Transformer-based architectures augmented by deterministic preprocessors of the input and data augmentation techniques.
> Concerning those preprocessing SMILES into graphs and using GNNs, we note that we could not effectively encode a large stoichiometric coefficient if not by cloning the input graph, yielding the same issues discussed above. We will discuss this in the main paper in detail.
> Furthermore, these pre-training techniques are shown to be insufficient for reaching an effective reasoning [1]. We argue that more profound innovations must happen at the architectural level of these models in order to solve the reasoning tasks.
> Lastly, note that ChemAlgebra already offers a more meaningful (for the purpose of our reasoning task) form of augmentation during training via different rebalancing of the reactions.
>
> Bibliography:
> [1] - Helwe, C., Clavel, C., & Suchanek, F. M. (2021, June). Reasoning with transformer-based models: Deep learning, but shallow reasoning. In 3rd Conference on Automated Knowledge Base Construction.
>
> > Absence of a version suitable for being used with large language models:
>
> We agree with the reviewer that this would be a very interesting direction! We are actually thinking of having a follow up work when we measure how large pre-trained language models can answer complex reasoning scenarios. However, this would require some careful engineering of the prompts to observe some reasonable performance.
> If the reviewer has a simple way to do it for ChemAlgebra, we are willing to run the experiment.
>
>
> > The Type 1 and Type 2 corruptions possibly alter the labels: It appear that both type 1 and type 2 corruptions could lead to a change in the labels. Here are some examples I can think of:
> Type 1: Take the following equation that shows a molecule of sugar being burnt: "C6H12O6+6O2→6CO2+6H2O". Doubling the coefficients on the left hand side, one could obtain the labels "12CO2+12H2O", or "6CO2+6H2O + C6H12O6 + 6O2" (the latter corresponds to only one of the sugar molecules being burnt, and the other remaining intact). Is one of the labels incorrect? Why?
>
>
> As usual in chemistry, we assume that whenever the reactants are enough to trigger a reaction, they will do so and transform into products. Hence there are no molecules in the products that can cancel out with those in the reactants. In yourexample, the correct label in the dataset would be “12CO2+12H2O".
>
>
>
> > Type 2 (This is the corruption used in section 3.2): Consider the salt equation "HCL + NaOH -> NaCl + H2O". Add NH4OH on the left hand side to obtain "HCL + NaOH + NH4OH". Since NH4Cl is a viable molecule, the label to this input could both be "NaCl + H2O + NH4OH" or "NH4Cl + HCL + H2O". Is one of the labels incorrect? Why?
>
> In type 2 augmentations, we only add a molecule chosen from the set of molecules that are already present in the reaction. In the example, a type 2 augmentation would only add a molecule chosen from {HCL, NaOH, NaCl, H2O}. This was done to mitigate the problem highlighted by the reviewer. We will explicitly include this clarification in Appendix C.

---

> > ### Author Response · Authors · 2022-11-14
> > **Response to reviewer 3JTU (2/3)**
> >
> > > Training details missing: Details on how the models used to produce (including tuning details) Table 2 and 4 appear to be missing. (Most of the details for Table 4 are provided, though there still remain key design choices, like the loss function used)
> >
> > We apologize if some of the fundamental details have gone missing.  Our loss function is the cross-entropy between the predicted and the target strings. We will include this additional training detail in the appendix. We are eager to include any additional information that the reviewer thinks is relevant for reproducibility. We remark that our code and datasets (and hence details) are present in the uploaded and anonymized repository specified in our reproducibility statement in page 10.
> >
> >
> > > The OOD split of T1 predictions seem a bit problematic:
> >
> > We agree that this is the hardest part of the benchmark, and as discussed on page 8 requires additional external arithmetic knowledge. We introduced it explicitly to stimulate future research to inject external knowledge or ways to bind symbols to concepts. We will make this clear in the main text.
> >
> >
> > > Why use a separate dataset (not ChemAlgebra) for Table 2?
> >
> > As discussed before, we had to use a separate dataset for Table 2 because the evaluated models do not have a way to explicitly represent stoichiometry. In practice, we had to replicate the molecules by copying them as many times as dictated by the stoichiometric coefficients. This results in hitting some practical limitations of these models, especially regarding maximum sequence length, which generally resulted in being able to perform only a more limited amount of stoichiometric augmentations than in ChemAlgebra.
> >
> > > How rule-based is the chemistry knowledge needed to perform perfectly?
> >
> > Some works try to encode chemical reactions mechanisms as electron movements, resulting in a sequence of bond-forming and bond-breaking actions between adjacent atoms. However, to the best of our chemical knowledge, this is still a simplification of the actual physical mechanisms involved, and we are not sure that this abstraction can be applied to all possible kinds of reactions.
> >
> > > Is natural language prone to shortcuts: In the 3rd paragraph of the Introduction, there's the sentence "Natural language, despite its flexibility, is[...] prone to shortcuts." Do you mind giving an example for what a shortcut for natural language would be?
> >
> > A shortcut is a simpler function that approximates well the true task on the training distribution, but not on a OOD distribution. An interesting example of shortcut in natural language is reported in [2], section 2.1. When given in input the sentence “Marcel Oopa died in the city of [MASK]”, a pre-trained BERT model was able to correctly answer “Paris”. However, when given in input the negated sentence  “Marcel Oopa did not die in the city of [MASK]”, the answer was still “Paris”. It is clear that the model here just learned a simple statistical correlation between the words “Marcel Oops”, “die”, “city” and “Paris”, instead of actually understanding the meaning of the question.
> >
> >
> > For more examples of shortcuts, please refer to the very interesting works of: [2, 3].
> > [2] - Helwe, C., Clavel, C., & Suchanek, F. M. (2021, June). Reasoning with transformer-based models: Deep learning, but shallow reasoning. In 3rd Conference on Automated Knowledge Base Construction.
> > [3] -  Geirhos, R., Jacobsen, J. H., Michaelis, C., Zemel, R., Brendel, W., Bethge, M., & Wichmann, F. A. (2020). Shortcut learning in deep neural networks. Nature Machine Intelligence, 2(11), 665-673.
> >
> >
> > > Loss function: What loss function did you use to train the transformer on ChemAlgebra?
> >
> > As loss function, we used the cross-entropy between the predicted and the target strings, in line with toothier state-of-the-art-models such as the Molecular Transformer or the Chemformer. We will include this information in the appendix.
> >
> > > Soundness of the procedure to miss omitted molecules:
> >
> > The re-balancing of USPTO reactions was done with the aid of the RDKit tool. We added the missing byproduct only if they formed a real-world molecule, recognized by RDKit. Combined with the fact that the starting reactions were taken from the real-word USPTO dataset
> >
> >
> > > Tokenization: How are the equations tokenized?
> >
> > In general, we tokenize the equations character-wise, except when an atom name was composed by more than one character. Hence, for example, the gold atom “Au” will be represented by a single token. Even in this case, we aligned ourselves with the other state-of-the-art models. Please refer to the source code for the exact tokenization algorithm (all the details are contained in the Tokenizer class of data.py).

---

> > > ### Author Response · Authors · 2022-11-14
> > > **Response to reviewer 3JTU (3/3)**
> > >
> > > > CLARITY: The paper is well-written throughout and is structured well. That being said, it still takes a couple of pages to finally see how the inputs and outputs look like for the chemical reaction prediction task. This can easily be remedied by either adding a short paragraph in the introduction, or perhaps adding a figure early on that demonstrates how input-output pairs look like in ChemAlgebra.
> > >
> > > Appendix G contains some samples taken from the different variants of the dataset. We will add a picture at the beginning of the paper, in compatibility with page limit restrictions.
> > >
> > > > Potential typo: The first paragraph of Section 3.1 appear to refer to two types of USPTO-MIT dataset, one that cites Lowe (2012), the another that cites Jin et al. (2017). Just wanted to check whether this is a typo and the first one is meant to be just called USPTO?
> > >
> > > Lowe (2012) is the first proposer of the USPTO dataset, while Jin et al. (2017) introduced the polished version USPTO-MIT. Many thanks for pointing that out, we will correct the typo in the paper.
> > >
> > >
> > > > REPRODUCIBILITY: There doesn't seem to be enough details to retrain the models used to report the main performance results. Also, the ChemAlgebra dataset also don't appear to be linked in the submission (is there a way to look at the samples during review?)
> > >
> > > We reported the dataset, the codes used to train the models and the script used to compute the results at the following link:
> > >
> > > https://anonymous.4open.science/r/ChemAlgebra
> > >
> > >  The same link is also included in the “Reproducibility statement” section at the end of the paper. We are eager to add to the appendices the additional training details that the reviewer deems to be relevant.

---

> > > > ### Comment · Reviewer_3JTU · 2022-11-15
> > > > **Thank you for your response.**
> > > >
> > > >
> > > >
> > > >
> > > > Thank you for your response and clarifications.
> > > >
> > > > My concerns related to 1) SOTA not being evaluated on ChemAlgebra, 2) the task not being directly suitable for LLMs, 3) additional training details, 4) and some others are addressed.
> > > >
> > > > Here are some remaining questions/concerns.
> > > >
> > > > I have some follow-up questions:
> > > >
> > > > **Non-uniqueness of answers:** Could you elaborate on this a bit more? Isn’t it true that the same reactants can produce different outputs depending on the physical properties of the environment in which the reaction is taking place? I remember this to be the case especially for organic chemistry, if my memory serves me well. Therefore, it appears to me that the non-uniqueness issue still could be a problem.
> > > >
> > > > **Additional chemistry knowledge required:** How much can one use pure reasoning to solve the problems described in ChemAlgebra? Based on the author responses, it looks more likely that any model that performs well on this benchmark will have to learn/memorize a significant body of chemistry facts. I’m not quite sure if this is an issue, but I believe this component of the task is orthogonal to evaluating reasoning. Another issue is that this fact might make model failures to be less interpretable. For example, how would one recognize if the model failed due to not having learned a chemistry fact, as opposed to having failed at a reasoning step?
> > > >
> > > > **Requirement of algebraic knowledge:** I believe that the issue regarding the OOD shifts being too difficult/impossible due to unseen digits being present in the equation is still present. Perhaps the authors could search for alternative distribution shifts that don’t require presenting the model with unseen digits? (I acknowledge that this is impossible to do within the rebuttal process)
> > > >
> > > > Lastly, it looks like this sentence in your response is incomplete: “Soundness of the procedure to miss omitted molecules:” Do you mind writing the completed sentence?
> > > >
> > > > All these being said, I still think that balancing stoichiometric equations equations is an interesting reasoning task that warrants attention and a standardized dataset, so the current submission is addressing a gap in literature.

---

> > > > > ### Author Response · Authors · 2022-11-18
> > > > > **We thank the reviewer for the insighful discussion**
> > > > >
> > > > > > Isn’t it true that the same reactants can produce different outputs depending on the physical properties of the environment in which the reaction is taking place?
> > > > >
> > > > > We agree that chemical reactions in the real-world can yield ambiguous products. We build our benchmark on top of existing datasets such as USPTO, and therefore inherit all their limitations. We just note that ambiguous reactions do not seem to be a big problem for current state-of-the-art Transformer-based models, which achieve more than 90% accuracy on USPTO.
> > > > >
> > > > > > How much can one use pure reasoning to solve the problems described in ChemAlgebra?
> > > > >
> > > > > We agree with the reviewer that solving ChemAlgebra requires both learning and reasoning and we cannot remove the learning component as a whole. See also our answer to Reviewer xL5G.
> > > > >
> > > > > > how would one recognize if the model failed due to not having learned a chemistry fact, as opposed to having failed at a reasoning step?
> > > > >
> > > > > To check for the failure on the learning side, we would check if the model is able to correctly predict the original (but balanced) reactions from USPTO.
> > > > > To check if the failure is on the reasoning side, instead we expect the model to correctly predict the original USPTO reaction but to fail to generalize to a Type 1 or Type 2 variant of the same reaction. Or, to increase the complexity of the reasoning task, if the model is able to predict one Type 1 reaction with a X-times multiplier for the stoichiometric coefficients, but not able to generalize over the (X+1)-times variant. This is the idea behind the design of the out-of-distribution and cross-distribution test sets.
> > > > >
> > > > >
> > > > > > Perhaps the authors could search for alternative distribution shifts that don’t require presenting the model with unseen digits?
> > > > >
> > > > > We point out that distribution shifts already happen in our cross-distribution setting: there, the model does not observe the one test reaction with the same stoichiometric coefficients at training time. Hence, there is already a shift from X to X+k molecules (or vice-versa).
> > > > > We agree that identifying different shifts would be interesting, and worth investigating in a follow-up.
> > > > >
> > > > > > Lastly, it looks like this sentence in your response is incomplete: “Soundness of the procedure to miss omitted molecules:” Do you mind writing the completed sentence?
> > > > >
> > > > > We apologize for the incomplete sentence. Here we report the full answer:
> > > > >
> > > > > The re-balancing of USPTO reactions was done with the aid of the RDKit tool. We added the missing byproduct only if they formed a real-world molecule, recognized by RDKit. Combined with the fact that the starting reactions were taken from the real-word USPTO dataset, this procedure should guarantee that the re-balanced reactions obtained are sound. We empirically observed that this procedure was mostly successful when the omitted molecules where simple byproducts such as “H2O” or “HCl”, while RDkit had a much harder time to recover complex molecules, or multiple ones from the same bag of atoms (the latter was the case when we tested this procedure on the final ChemAlgebra datasets). So we think that this re-balancing procedure is quite conservative, preferring to not rebalance a reaction if that could introduce potential soundness problems.
> > > > >
> > > > > > All these being said, I still think that balancing stoichiometric equations equations is an interesting reasoning task that warrants attention and a standardized dataset, so the current submission is addressing a gap in literature.
> > > > >
> > > > > We thank the reviewer for recognizing the novelty and potential impact of this work and, above all, for this engaging discussion which is helping us clarifying and improving the scope and goal of our work!

---

> > > > > > ### Comment · Reviewer_3JTU · 2022-11-18
> > > > > > **Thanks again**
> > > > > >
> > > > > > Thank you for your responses. I'll consider your answers and update my score after the reviewer discussion.
> > > > > >
> > > > > > I have one final thought, which might be considered as external to this review: Did you consider a different version of the task where all that the model needs to do is to predict the reaction coefficients, instead of that *and* the output molecules? The input to model could looks something like:
> > > > > >  [...inputs...] -> [1. output molecule 1] [2. output molecule 2] ... [n. output molecule n] [SEP] [coeff 1] [coeff 2] ...[coeff n]
> > > > > > The model would be trained to predict [coeff 1] [coeff 2] ...[coeff n] given the rest.
> > > > > >
> > > > > > This way the chemistry knowledge required for this task would be removed and pure reasoning would be enough. Additionally, this proxy task could be useful to measure how much of a role chemistry knowledge plays in this task: the difference between the performance on the full task (predict coeffs as well as the moledules) and the performance on only predicting the coefficients would give an estimate of what percent of the errors are due to lack of chemistry knowledge.
> > > > > >
> > > > > > Do you think this makes sense? (To be clear, I'm definitely not asking the reviewers to incorporate this feedback in the paper!)

---

> > > > > > > ### Author Response · Authors · 2022-11-18
> > > > > > > **Very interesting line of future investigations**
> > > > > > >
> > > > > > > We agree it can be interesting investigating if models can simply "count". However, the model would still need some chemical knowledge to parse SMILES. For that purpose, however, predicting raw chemical formulas could be easier. We will definitely include these investigations in the second version of ChemAlgebra, after we release this initial version. Thank you for the suggestion, we really enjoyed this discussion!

---

> > > > > > > > ### Comment · Reviewer_3JTU · 2022-11-26
> > > > > > > > **Thanks again**
> > > > > > > >
> > > > > > > > Thanks again.
> > > > > > > >
> > > > > > > > I'll improve my score in response to the rebuttal, pending reviewer discussion. Though some of my concerns remain (especially regarding unseen digits), I still believe that the task (perhaps with some small but important modifications) is very suitable for evaluating reasoning and hence is promising.

---

### Official Review · Reviewer_xL5G · 2022-10-24

**Confidence:** 3
**Correctness:** 2
**Technical Novelty And Significance:** 1
**Empirical Novelty And Significance:** 2
**Recommendation:** 5

**Clarity, Quality, Novelty And Reproducibility:**

Clarity: Needs improvement (O1, O2, O3), and the expected use of this work (or the target community) should be made more explicit (W2, W3).



Quality: On the high level the setup is reasonable, but I have doubts about many details (see the "Strengths and Weaknesses" section), and I'm not convinced the comparisons made in this work are very useful.



Reproducibility: Seems OK, although some of the numbers look confusing (O2), making me think that some of the details were either not provided or I didn't fully understand them.

**Strength And Weaknesses:**

=== Strengths ===

(S1): The paper considers an important problem of chemical reaction prediction, and highlights limitations of current models (although these limitations were known before this work).



=== Weaknesses ===

(W1): I am not convinced that the setup and evaluations done in this work are completely fair, and if it's realistic to expect the models to perform well in these settings.

- As far as I understand, in Section 3.3 the authors train the models on the original USPTO dataset, and then evaluate on the newly introduced variants such as USPTO-BAL, reporting very low results. I'm not sure if it's realistic to expect the models to transfer here. As the author note, in USPTO some side products are omitted, hence the models are taught to omit side products; in a way, not making the reactions balanced is part of the task. The balanced reactions on the other hand teach the model to make different assumptions. Hence I'm not sure how the models could generalize from one to the other, if part of the task specification is contradictory.

- Moreover, the authors refer to going from USPTO to USPTO-T{1,2} as a "small distribution shift", but I don't think this is accurate. Going from no repeated reactants to having repeats seems like a large shift, and I'm again not sure if it's realistic to expect the models to generalize. It would be a small shift to go from reactants repeated at most X times (for X > 1) to reactants repeated at most X' times (for X' > X).

(W2): The language of the paper seems slightly misleading, as it seems to suggest not being able to capture stochiometry is a general problem that all reaction models would have. However, I think the discussion applies mostly to Transformer-based models (or other unconstrained ones). Models that predict the reactant-product transformation as a edit may capture stochiometry much better, and some may even satisfy it by design; for example, if the reactions are balanced and have a full atom mapping, and one trains a reaction model based on templates or MEGAN [1], only allowing bond changes, then atom counts would be preserved by design. Therefore, it's not completely clear that addressing the highlighted failures of Transformers is the right approach to getting generalizable reaction models, or at the very least it's not the only viable one. While the authors do discuss non-Transformer models in Appendix A, I'm not fully satisfied with the discussion there either; e.g. the authors mention several models, and summarize things by saying that these models are "restrained to a particular class of reactions and cannot easily generalize to new types", which I don't think is the case for MEGAN (which the authors do mention).

(W3): The paper seems to pitch ChemAlgebra as a general benchmark that will help us improve the reasoning capabilities of DL models more broadly. However, all tasks in ChemAlgebra require not only learning generalizable chemical rules (such as stochiometry), but also predicting the actual reaction mechanism, and the latter is a harder task (since, as I mentioned in (W1), not breaking these simple rules can be baked in into the models by design, and one cannot bake in the ability to correctly predict reactions). Hence, I'm not sure improvements on ChemAlgebra would actually mean better reasoning, as opposed to just better reaction prediction. I think the paper should position itself here more explicitly: either as something that is supposed to be useful for the ML4Chem community (i.e. people working on reaction prediction) or something useful for the broad ML community (i.e. people that focus on Transformers but likely have no interest in chemistry).

=== Other comments ===

(O1): "To disentangle the reasoning task of predicting graph structures from that of counting molecule multiplicities, we devise a second, simpler, representation where molecules in the reactions are encoded as raw chemical formulas." - Are these tasks really disentangled here, given that even in the chemical formula representation one still has to predict the actual reaction mechanism (which the authors note is actually quite hard in this representation too)? I think fully disentangling these tasks would be e.g. giving the model both reactants and products, and asking only for the stochiometric coefficients. At that point however this becomes a toy task, and the fact that it came from a real task in chemistry is unimportant.

(O2): Is each of the 8 tasks in ChemAlgebra comprised of its own train set and test set? If yes, I don't understand why the in-distribution results in Table 4 are so low (close to 0 for the SMILES representation). If this is in-distribution, shouldn't the models do similarly well to how they do on normal USPTO?

(O3): I don't understand the last two sentences of Section 4.1, which talk about the formula-based tasks still being challenging. Why would we expect reaction prediction in the formula space to be easy? The authors themselves note that predicting the reaction mechanism from formulas themselves is actually harder.

=== Nitpicks ===

Here I include some final nitpicks, which did not affect my score; they are here just to help improve the paper.

- "These impressive performance" - maybe "this"?

- "This constraint over atoms of the molecules undergoes the true chemical mechanism behind reactions" - maybe "undergoes" -> "underpins"?

- "In fact, predicted product molecules might sport" - not sure what "sport" means here

- "apply it too all molecules" - presumably "too" -> "to"



=== References ===

[1] Sacha et al, "Molecule Edit Graph Attention Network: Modeling Chemical Reactions as Sequences of Graph Edits"

**Summary Of The Paper:**

This paper analyses the performance of Transformers on reaction prediction, highlighting that these models break stochiometric rules and fail to follow "common sense". In light of this, the authors propose a benchmark – called ChemAlgebra – used to evaluate the ability of Transformer models to learn these generalizable chemical rules such as stochiometry.

**Summary Of The Review:**

Overall, I'm not convinced the comparisons made in this work are fully fair, or that they give useful signal, as in many cases the models seem to be evaluated too out-of-distribution for it to be meaningful. Moreover, I feel like some of the discussion in the paper may be misleading, and it's not clear which community is this work trying to benefit. Hence, for my initial evaluation I lean towards rejection.

=== Update after author response ===

The author response clarified a few things, and the framing was changed to pivot more on the side of reasoning with Transformers, and less on usefulness for real-world reaction prediction. However, I'm still not very convinced about the OOD setups being sensible here (in particular, I share the concern of Reviewer 3JTU around some tokens not being seen during training), and about the signals from doing "good reasoning" and "good reaction prediction" being conflated. I raised my score to 5 to reflect the updates, but that already feels to be like a stretch; I would still vote for rejection.

If the paper gets rejected and resubmitted to the next venue, I think the suggestion from Reviewer 3JTU to include a task where the model only needs to predict the coefficients is very valuable.

---

> ### Author Response · Authors · 2022-11-14
> **Response to reviewer xL5G (1/2)**
>
> We thank the reviewer for the great attention they put for reviewing the paper. We will try to address the individual concerns in the following.
>
> > I'm not sure if it's realistic to expect the models to transfer here.
>
> We agree that there is a distribution shift. Our major point here is that in order to systematically generalize, i.e., to generalize despite distribution shifts, Transformer models should be equipped with robust reasoning routines. The fact that these models fail on these USPTO variants, highlights they are not able to and further progress is needed in this direction. ChemAlgebra is an initial stepping stone to measure such a progress.
>
> Even failures over the  BAL, T1 and T2 variations are worth highlighting and measuring.
> As long as we will use simple datasets such as USPTO for training our models, it will be difficult to achieve systematic generalization and, as a consequence, to develop models capable of actual robust predictions.
>
> > as a "small distribution shift", but I don't think this is accurate. Going from no repeated reactants to having repeats seems like a large shift, and I'm again not sure if it's realistic to expect the models to generalize.
>
> We agree that the sentence can be ambiguous. We will rephrase it in the paper to say that the shift still uses the same molecules and reaction mechanisms of USPTO. Only for the T2 variant of BAL where we add a single molecule, the shift is minimal.
>
> > It would be a small shift to go from reactants repeated at most X times (for X > 1) to reactants repeated at most X' times (for X' > X).
>
> This kind of reactions appear in the full ChemAlgebra benchmarks in the in- and cross-distribution settings where models do see reactions at training time with certain multiplicities which are increased at test time.
>
> > Models that predict the reactant-product transformation as a edit may capture stochiometry much better,
>
> We focused on Transformer-based models because, to the best of our knowledge, this approach is currently reporting the highest accuracy in chemical reaction prediction. In addition, Transformer-based models are also already used for reasoning tasks, so a comparison with them is more relevant for the reasoning community.
>
> Please note that the MEGAN model reports a lower top-1 accuracy on forward reaction prediction than the models discussed in Table 2 of our paper. But it is true that MEGAN seems to be able to model different classes of reactions. We will correct Appendix A accordingly: thank you for pointing this out.
>
> We indeed agree that it is challenging for Transformers to be able to solve the balanced chemical reaction prediction task perfectly, and that more structured approaches (such as the one of MEGAN) will be needed. We hope that ChemAlgebra can be a useful way for spurring the community to devise innovative and effective solutions for the task, and to give higher visibility to more grounded approaches such as MEGAN.
>
> > I think the paper should position itself here more explicitly: either as something that is supposed to be useful for the ML4Chem community (i.e. people working on reaction prediction) or something useful for the broad ML community (i.e. people that focus on Transformers but likely have no interest in chemistry).
>
> The main focus of this paper is to propose a benchmark for evaluating the algebraic reasoning abilities of deep learning models, so the intended use is mainly for the broad ML community. The chemical reaction prediction task is a proxy task for measuring algebraic reasoning. We will stress this in the introduction.
>
> > I'm not sure improvements on ChemAlgebra would actually mean better reasoning, as opposed to just better reaction prediction.
>
> In order to score well on ChemAlgebra, a model needs to solve both the stoichiometry and the reaction mechanism. Predicting the reaction mechanism alone is not enough for achieving a satisfactory performance on ChemAlgebra. For example, if the label is “{3}H2O”, while the model predicts only one “H2O”, thus correctly predicting the mechanisms but failing on balancing the stoichiometry, the model will get an accuracy of only 33%.
> Nevertheless, we think that there can be some usefulness in applying ChemAlgebra to reaction prediction. For example, while many rules can be baked into the models by design, it can still be useful for a model to infer its own rules based on the task.
> Concerning reasoning alone, ChemAlgebra can propel research over Transformers that consistently struggle to count and reason about multiplicity in natural language and computer vision e.g., [1]
>
> [1] Yasaman Razeghi, Robert L Logan IV, Matt Gardner, and Sameer Singh. Impact of pretraining term frequencies on few-shot reasoning. arXiv preprint arXiv:2202.07206, 2022

---

> > ### Author Response · Authors · 2022-11-14
> > **Response to reviewer xL5G (2/2)**
> >
> > > Are these tasks really disentangled here,
> >
> > We agree they cannot be, in general, completely disentangled. We will change the sentence to “To better understand how the effect of the reasoning task over performance”.
> > We note it is not possible to reconstruct the molecular graph starting from the molecular formula, while it is possible to do so starting with SMILES. While some degree of reaction mechanism prediction is still involved with formulas, we think that the more abstract representations only allows the model to infer more “high-level pseudo-mechanisms”, instead of the low-level bond-forming and bond-breaking mechanisms. However, we agree that the two tasks cannot be deemed completely disentangled.
> >
> >
> > > why the in-distribution results in Table 4 are so low (close to 0 for the SMILES representation). If this is in-distribution, shouldn't the models do similarly well to how they do on normal USPTO?
> >
> > We confirm that each task has its own training, validation and test sets. Please note that the ChemAlgebra tasks are based on the BAL version of USPTO. This can be one of the reasons for the low performance. Furthermore, the EM metric requires to perfectly predict both the products and the stoichiometry, so it is very difficult to reach a good score with it. Other metrics are less strict and therefore yield higher performance.
> >
> > > Why would we expect reaction prediction in the formula space to be easy?
> >
> > We will rephrase the sentence to highlight that predicting multiplicities alone should be easier from a syntactical point of view when we deal with formulas. Thank you for pointing this out.

---

> ### Author Response · Authors · 2022-11-18
> **We hope we clarified all the reviewer's concerns**
>
> We hope that our response was able to clarify all the concerns raised by the reviewer. We kindly ask the reviewer to let us know if they have some additional questions that they would like to be addressed.
>
> We look forward to engage in a fruitful discussion!

---

> > ### Comment · Reviewer_xL5G · 2022-11-27
> > **Response to authors**
> >
> > Thanks for a detailed response! I reviewed your answers, and also the discussions with the other reviewers. I think the current framing that leans more on the side of reasoning with Transformers is better.
> >
> > That being said, I'm still not convinced. In particular, I worry about the OOD setups not being sensible (I agree with the concern of Reviewer 3JTU around some tokens not being seen during training), and the signals from doing "good reasoning" and "good reaction prediction" seem too conflated to draw good insights from the results (the suggestion from Reviewer 3JTU to include a task where the model only needs to predict the coefficients would help with this).
> >
> > I raised my score to 5 to reflect the updates, but I still lean towards rejection. I will continue the discussion with the other reviewers and the AC.

---

### Official Review · Reviewer_gwmV · 2022-10-27

**Confidence:** 3
**Correctness:** 3
**Technical Novelty And Significance:** 2
**Empirical Novelty And Significance:** 3
**Recommendation:** 5

**Clarity, Quality, Novelty And Reproducibility:**

This paper is clearly written with high quality. The work is novel and reproducible.

**Strength And Weaknesses:**

Strength:
- This paper is well-written and easy to follow.
- The formulation of the task is novel. It combines the flavor of algebraic reasoning from math QA and the flavor of learning chemical reactions. This would be an interesting task for researchers in both domains.
- Several variations of the dataset were proposed (BAL, T1, T2, and other 8 variations) to help researchers understand more about the distribution of the datasets, the difficulties of each dataset, and what aspects of the datasets are more challenging.
- Recent chemical reaction baselines are included in the proposed benchmark.

Weaknesses:
- Despite the novelty and interesting side of the proposed task, the usefulness of such a task in real-world applications is still less certain. The current reaction prediction model can be used to predict the main product, and with the re-balancing techniques, we can derive the side-product as well as the multiplicities of the product molecules. The paper needs to justify more why such a task would be meaningful in real-life.
- Similar to the first argument, such a main product prediction + re-balancing method can be added as one of the baselines in the benchmarks. The method is straightforward and should be included as a counterpart of the end-to-end baselines.


**Summary Of The Paper:**

This paper proposed a new algebraic reasoning task and benchmark on chemical reactions. The new task is a modified version of chemical reaction prediction. Instead of only predicting the main product in the reaction, we need to predict all the products, including the by-products and the multiplicities of the product molecules. The proposed benchmark was constructed out of the USPTO-MIT dataset by doing a re-balancing on the product molecules. To establish the baselines, molecular transformers and graph neural networks were tested on the main benchmark and its variations. The task was shown to be a challenging one for the existing baselines from the experiment results.

**Summary Of The Review:**

In general, the paper is well-written. The proposed task is novel and interesting for both math QA and ML for science communities. However, the paper needs to justify more on its impact on real-world applications and include a straight-forward but not end-to-end baseline.

---

> ### Author Response · Authors · 2022-11-14
> **Response to reviewer gwmV**
>
> We thank the reviewer for the overall positive assessment of our work and the recognition of its novelty and potential interest for the community. We elaborate below on specific questions and we will updated the main paper accordingly.
>
> > the usefulness of such a task in real-world applications is still less certain.
>
> We remark that our main contribution is a challenging benchmark for machine reasoning and not a benchmark for computational chemistry per se. Its use is therefore to be found in the ability to challenge the current state-of-the art of neural reasoners and help compare them.
> To see why such a benchmark is needed, consider one of the most prominent benchmark in neuro-symbolic reasoning these days: MNIST digit addition, as introduced in [1]
>
> There, a reasoner has to predict the sum of some integers encoded as MNIST images.
> The purpose of such a benchmark is not to advance algorithmic addition nor digit classification, but show how this is challenges for image classifiers and requires some reasoning capabilities.
> In the same way, ChemAlgebra raises the bar and offers an inherently more challenging task than MNIST addition, as it operates on bags of graphs and requires reasoning over the multiplicities of node types in these graphs, as discussed in the paper.
>
>
> [1] - Manhaeve et al., “Deepproblog: Neural probabilistic logic programming”, NeurIPS, 2019
>
> > such a main product prediction + re-balancing method can be added as one of the baselines in the benchmarks
>
> We first point out that given an unbalanced set of predicted products, to yield a balanced reaction one might need to add molecules to both products and reactants (i.e.,). See for example, the third reaction in Fig 1. This means that the proposed baseline would operate on a different task, as in ours the set of reactants and reagents is given and fixed.
>
> > By running it on the predictions of Transformer….of USPTO-BAL we find that only….
>
> Lastly,  note that the re-balancing technique we used is a simple heuristics: we were only able to re-balance about 45% of original USPTO. Sometimes the collection of “spare atoms” is ambiguous, and multiple molecules could be inferred.  It seems unrealistic to rely on that in real life reaction predictions.
>
> As the reviewer suggested, we also tried to use the re-balancing heuristic on ChemAlgebra, comparing the bag of atoms in the prediction with the one in the initial reactants. We computed the bag of atoms that were missing from the prediction, and tried to use RDKit to infer the missing molecules. We found that it was not possible to re-balance any additional reactions, making RDKit not suitable as a rebalancing tool in the presence of complex stoichiometric reactions.

---

> > ### Comment · Reviewer_gwmV · 2022-11-23
> > **Thanks for the response**
> >
> > Thank you for the detailed response! However, I did not find the response convincing for both of my concerns.
> >
> > "Its use is therefore to be found in the ability to challenge the current state-of-the art of neural reasoners and help compare them. "
> >
> > If the sole purpose of this benchmark is to challenge existing models, this benchmark would be much less useful and meaningful. I am also not sure how can such a benchmark improve the state-of-the-art neural reasoners because it is hard to come up with a general neural reasoner that can solve such a task without leveraging domain knowledge.
> >
> > "We first point out that given an unbalanced set of predicted products, to yield a balanced reaction one might need to add molecules to both products and reactants"
> >
> > I don't see why this is an obstacle to using such a baseline in the experiments. As a modification, we can train a Transformer to predict all the products including the added molecules (without multiplicities), and then rebalance them.

---

> ### Author Response · Authors · 2022-11-18
> **We hope we clarified all the reviewer's concerns**
>
> We hope that our response was able to clarify all the concerns raised by the reviewer.
> We kindly ask the reviewer to let us know if they have some additional questions that they would like to be addressed.
>
> We look forward to engage in a fruitful discussion!

---

### Official Review · Reviewer_7VBJ · 2022-11-19

**Confidence:** 5
**Correctness:** 3
**Technical Novelty And Significance:** 3
**Empirical Novelty And Significance:** 3
**Recommendation:** 6

**Clarity, Quality, Novelty And Reproducibility:**

clarity:
it is not clear whether the baselines are actually retrained on the new benchmark datasets, or whether checkpoint trained on the unbalanced USPTO is used. If the latter is the true, it's no surprise that the models perform poorly (because the main assumption for MLE is that training and test set should be sampled from the same distribution). the authors need to clarify this.

quality:
the task of predicting molecular formulas is very hard, since it is ambiguous. it's also not clear to me why this is relevant to chemistry.

**Strength And Weaknesses:**

strengths
- critical evaluation of the supposed state of the art in reaction prediction

weaknesses
- somewhat irrelevant to real-world reaction prediction
- unclear whether baselines are run in a fair way
- important baselines operating on graphs (e.g. MEGAN) missing

**Summary Of The Paper:**

A new benchmark for chemical prediction is proposed, accounts for stoichiometry. this is supposedly harder than existing benchmarks.
transformer baselines are evaluated on the task, and don't seem to perform well.

**Summary Of The Review:**

interesting benchmark, questions remain.

---

> ### Author Response · Authors · 2022-11-20
> **Response to Reviewer 7VBJ**
>
> We thank the reviewer for recognizing that the benchmark is interesting. We point out that our focus is on reasoning, please see our answers to the other reviewers on this point.
>
> >It's no surprise that the models trained on unbalanced USPTO perform poorly (because the main assumption for MLE is that training and test set should be sampled from the same distribution)
>
> We agree that there is a distribution shift. As stated in the introduction, we remark that in order to systematically generalize, i.e., to generalize despite distribution shifts, Transformer models should be equipped with robust reasoning routines. That is, they should be able to generalize over simple variants of the same task.
>
> The first step is to measure how much they do fail. What is interesting is that the drop in performance is very large. This highlights that further progress is needed in this direction. ChemAlgebra is an initial stepping stone to measure such progress.
> As long as we use simple datasets such as USPTO for training our models, and we stick to the i.i.d. assumption, it will be difficult to achieve systematic generalization and, as a consequence, to develop models capable of actual robust predictions.
>
> > the task of predicting molecular formulas is very hard, since it is ambiguous. it's also not clear to me why this is relevant to chemistry.
>
> We remark that our main contribution is a challenging benchmark for machine reasoning and not a benchmark for computational chemistry per se. Please see our answers to the other reviewers and the paragraph we introduced on “Limitations and Future directions” in the revised version of the paper.
>
> The impact of ChemAlgebra has to be understood in terms of the current benchmarks used to measure machine reasoning capabilities, e.g., in NLP where benchmakrs are prone to many ambiguities coming from natural language and offering statistical shortcuts (please see our answer to reviewer 3JTU) or in the neural-symbolic community, where benchmarks are too simplistic from a reasoning perspective and involve simple tasks such as performing addition over the class of images. Please see also our answer to reviewer gwmV.
>
> > important baselines operating on graphs (e.g. MEGAN) missing
>
> We focused on Transformer-based models because, to the best of our knowledge, this approach is currently reporting the highest accuracy in chemical reaction prediction. Please note that the MEGAN model reports a lower top-1 accuracy on forward reaction prediction than the models discussed in Table 2 of our paper. There are also practical issues with training GNN-based models on stoichiometric reactions (please see our response to reviewer 3JTU).
> We indeed agree that a pure Transformer architecture trained with MLE is unlikely to be able to solve the balanced chemical reaction prediction task, and that more structured approaches (such as the one of MEGAN) will be needed. We hope that ChemAlgebra can be a useful way for spurring the community to devise innovative and effective solutions for the task, and to give higher visibility to more grounded approaches such as MEGAN.

---

> > ### Comment · Reviewer_7VBJ · 2022-11-20
> > **thanks for the additional info.**
> >
> > > There are also practical issues with training GNN-based models on stoichiometric reactions (please see our response to reviewer 3JTU)
> >
> > I disagree with this: Actually a graph based model will be quite easy to use for stoichiometric reactions.
> >
> > However, in the updated version the authors argue more in the direction of transformers for reasoning tasks, rather than real world reaction prediction. With this framing, I am happy.

---

### Author Response · Authors · 2022-11-14
**General response to reviewers**

We thank the reviewers for their insightful suggestions and comments. Specifically, we are glad that they recognized the novelty (gwmV, Mbs4) and timeliness (3JTU) of the proposed benchmark, and deemed the use of balanced chemical reaction an interesting and good way for measuring algebraic reasoning (3JTU, Mbs4) as well as well-written and easy to follow (gwmV, 3JTU, Mbs4)


We are preparing a revised version of the manuscript that tries to include all the improvements that the reviewers suggested.
In the meantime, we answered the individual concerns of the reviewers in the review’s reply.
The main points are, in synthesis:
We remark that the main contribution of this paper is to provide a strong benchmark for machine algebraic reasoning and not for chemistry per se
- We provided additional justification for the real-world usefulness of the benchmark.
- We better explained some potentially ambiguous sentences in the paper.
- We discussed the inclusion of additional baselines results.
- We described in more detail some of the steps used for creating the benchmark.

We look forward to reviewers’ responses and to engage in a fruitful discussion!

---

### Author Response · Authors · 2022-11-15
**A revised version of the paper is available.**

Following the suggestions of the reviewers, we prepared a new version of the paper. In particular:

- We  rephrased the "small distribution shift" part, and added more justifications in Sec. 3.2.

- We changed the MEGAN discussion in appendix A.

- We reprased the part regarding disentanglement of reasoning tasks, after Eq. 6.

- We updated the discussion of results at the end of Sec. 4.1.

- We clarified the algorithm of appendix C.

- We added the loss function detail on appendix D.

- We added the out-of-distribution comment in Sec. 4.

- We added a footnote specifying why we chose Transformers as baseline, at the beginning of Sec. 4.1.

- We corrected several typos.

The revised parts are highlighted in purple color.
We thank again the reviewers for their insightful suggestions.

---

> ### Author Response · Authors · 2022-11-18
> **Added a "Limitations and Future Directions" section**
>
> Following the discussion with the reviewers, we updated the manuscript, adding a new "Limitations and future directions" section at the end of the paper. In that section, we better highlight the current limitations of our approach, and clarify the main goal and scope of this work.
>
> The revised parts are highlighted in purple color.
> We thank again the reviewers for having helped us improving the paper.

---

### Decision · Program_Chairs · 2023-01-20

**Decision:**

Reject

**Justification For Why Not Higher Score:**

As explained by reviewer 3JTU, there are fundamental design choices made in the task that is problematic as reasoning benchmark.

**Justification For Why Not Lower Score:**

N/A

**Metareview: Summary, Strengths And Weaknesses:**

This paper proposes a benchmark for measuring reasoning capabilities of deep models through predicting chemical reactions. The paper is well-written and easy to follow. It also nicely shows some of the limitations of transformers. However, the main shortcoming of the paper is that as a reasoning benchmark, it has issues that limits its effectiveness in measure the reasoning capabilities of the models. In my mind, the most important limitations are regarding the separation of chemistry knowledge from reasoning capabilities and the issue with unseen digits raised by reviewer 3JTU. Therefore, I'm recommending rejection. However, I suggest authors to take reviewers suggestions into account and resubmit.

**Summary Of Ac-Reviewer Meeting:**

3JTU:
1- It is not clear how much chemistry knowledge is needed to success in this task so this might not be an ideal benchmark for reasoning. For example, it is not clear if a human can be successful in this task.
2- In type 1 task, it is not clear if model could generalize to unseen digits.
xL5G:
1- Showing the limitations of transformer models is interesting.
2- Unfortunately, the paper is mixing reaction prediction task with reasoning and it doesn't seem to be clearly championing any one of them.
7VBJ: The main interesting finding is showing the limitations of transformers.